# VIDEO-SALMONN S: STREAMING AUDIO-VISUAL LLMS BEYOND LENGTH LIMITS VIA MEMORY

## ABSTRACT

Continuous, high-frame-rate, high-resolution processing of long video streams is critical for future AI agents, yet current video-understanding LLMs struggle to scale. Offline, fixed-frame-number methods require the stream length to adapt frame rates; streaming methods constrain memory by merging or discarding tokens, losing information. We propose video-SALMONN S, a streaming audio–visual LLM that, to our knowledge, is the first to process >**3-hour** videos at **1 FPS** and **360p resolution** under a fixed memory budget. Our model introduces (i) a test-time-training (TTT) memory module that continually updates token representations to capture long-range dependencies by replacing token merging, and (ii) a prompt-dependent memory reader that selectively retrieves context-relevant content from fixed-size memory. The TTT module is optimised with a Hessian-free conjugate-gradient procedure ($\text{TTT}_{\text{HF}}$) for efficient adaptation. On long-video benchmarks (Video-MME, LVBench, VideoEvalPro), video-SALMONN S sustains high-quality understanding on multi-hour videos with >**10k frames** and ∼**1M tokens**. Our 8B-parameter model achieves **74.2%** overall and **67.8%** on the Video-MME long split, outperforming both offline and streaming baselines.

## 1 INTRODUCTION

Processing video streams of any length at a fixed high frame rate and a decent resolution is one of the crucial abilities in future AI agents. Despite the rapid progress in state-of-the-art visual and audio-visual large language models (LLMs) (Li et al., 2024; Zhang et al., 2024d; Wang et al., 2024a; Lin et al., 2024a; Bai et al., 2025; Zhang et al., 2025a; Tang et al., 2025a; Sun et al., 2025), most of them only perform offline video understanding by setting a maximum number of input visual tokens to the Transformer, hence yielding a significant information loss for long videos. Some recent work explores token compression methods to achieve longer video understanding (Li et al., 2025b; Tan et al., 2024; Tang et al., 2025b; Chen et al., 2024a; Zhang et al., 2025c), especially ones using a prompt to achieve high compression ratios (Gao et al., 2024; Wang et al., 2025b).

However, since it is usually unknown how long the video can be or when the user prompt may come, it becomes impractical for those offline LLMs in real-world scenarios, as Transformers always have a limited attention span. Streaming video understanding models that process video at a fixed frame rate are then developed, mostly merging or discarding tokens to construct a fixed internal memory (Qian et al., 2025; Yang et al., 2025a; Song et al., 2024; Zhang et al., 2025b; Huang et al., 2025). These methods predominantly focus on short videos due to significant information loss when the video gets much longer, which often yields degraded performance compared to their offline counterparts. More recent work explores agentic methods by constructing an external database and retrieving from it (Long et al., 2025). While being able to handle longer videos, dedicated training data is required, and the external memory size grows linearly with the video length.

To overcome these limitations, we propose video-SALMONN S, a streaming audio-visual LLM capable of understanding >3-hour videos at 1 FPS and 360p resolution under a fixed memory budget. Our method introduces the following two key innovations in long-term memory design.

**Test-time training (TTT) for memory writing**: Instead of discarding or merging, we apply a lightweight TTT layer to visual tokens, encoding long-term context into TTT parameters. Unlike a single recurrent hidden state (Gu & Dao, 2023; Dao & Gu, 2024), the TTT layer stores history

in its parameters, enabling stronger long-term dependency modelling. To further enhance memorisation, we propose $TTT_{HF}$, which leverages Hessian-free optimisation via conjugate gradient (CG) (Martens, 2010b), efficiently minimising reconstruction loss with only a few test-time updates.

**Prompt-dependent memory reading**: Inspired by AdaReTaKe (Wang et al., 2025b), we design an attention-based retrieval mechanism to select relevant KV-cache entries of the LLM backbone conditioned on the prompt, allowing large fixed-size memory to be maintained in streaming mode.

We evaluate video-SALMONN S on both standard and extremely long video benchmarks, including LVBench (Wang et al., 2024b) and VideoEvalPro (Ma et al., 2025). Videos are processed at 1 frame per second (FPS) and 360p resolution, corresponding to around 10k frames and up to 1M tokens in extreme cases. Results show that video-SALMONN S not only outperforms offline counterparts on long videos, but also achieves state-of-the-art (SOTA) performance overall among 7B/8B models. In particular, our 8B audio-visual model achieves 74.2% overall accuracy on VideoMME, with 67.8% on the long-video partition. Main contributions are summarised as follows:

- We propose video-SALMONN S, the first audio-visual LLM that can understand >3-hour standard definition of 360p videos at 1 FPS. video-SALMONN S employs a novel streaming video understanding framework including a TTT-powered long-term memory module and a prompt-dependent information extraction mechanism.

- We propose $TTT_{HF}$, which uses the Hessian-free (HF) method to further boost the memory efficacy in video-SALMONN S. $TTT_{HF}$ helps the model to achieve better performance by achieving a better convergence of TTT reconstruction loss without harming the information flow from each TTT update through the TTT layer.

- video-SALMONN S achieves streaming understanding of video of multiple hours corresponding to around 1M visual tokens in total, and yields better performance compared to non-streaming models with a notable 74.2% accuracy on Video-MME.

## 2 RELATED WORK

### 2.1 LONG VIDEO UNDERSTANDING

The key challenge of long video understanding is efficient compression of information into a size that the LLM can process. A number of training-based methods (Li et al., 2025b; Zhang et al., 2024b; Shen et al., 2024; Shu et al., 2024; Liu et al., 2025a) have been proposed to reduce the number of visual tokens needed to represent each video frame. Some recent visual LLMs, such as Qwen2.5-VL (Bai et al., 2025), inherently contain token merging modules and a much longer context LLM backbone that are suitable for long video understanding. Meanwhile, training-free methods (Liu et al., 2025b; Zhang et al., 2024c; Yang et al., 2025a; Wang et al., 2024c; 2025b) are also investigated, which mainly select KV-cache in each Transformer block. Specifically, ReTaKe (Wang et al., 2024c) applies a dynamic key-value (KV) cache compression by selecting and keeping the important KV-pairs only based on their similarities. The follow-up work, AdaReTaKe (Wang et al., 2025b) applies prompt-based KV-Cache selection depending on the attention scores.

### 2.2 ONLINE VIDEO UNDERSTANDING AND TEST-TIME TRAINING

Online video understanding requires processing frames continuously in real time, where the stream length is unknown, precluding the offline practice of pre-selecting a fixed, uniformly sampled frame set. Models, therefore, must operate under a fixed memory budget to stay within the LLM's context window. Prior work follows two main directions. Token-reduction methods fix the number of visual tokens: MovieChat Song et al. (2024) merges tokens via similarity-based consolidation, and VideoLLMonline (Chen et al., 2024a) reduces each frame to about 10 tokens for efficiency. External-memory methods compress and organise visual tokens and retrieve the most relevant ones at query time: Flash-VStream (Zhang et al., 2024a) and Dispider (Qian et al., 2025) fuse retrieved visual tokens with text tokens before feeding them into the multimodal LLM. A complementary line targets KV-cache compression and retrieval: ReKV (Di et al., 2025) offloads layer-wise caches to external storage for on-demand fetching; Ning et al. (2025) compresses the KV cache to cut memory and accelerate question answering (QA) relative to ReKV; StreamMem and InfiniPot-V (Kim

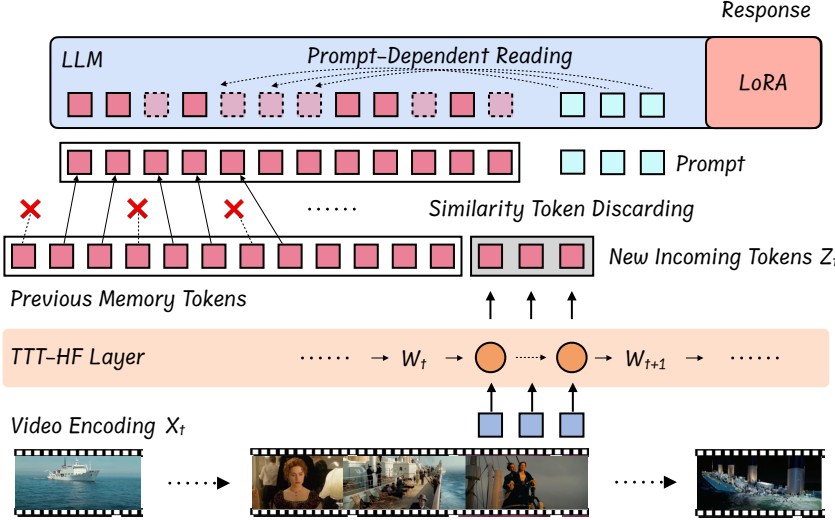

Figure 1: Overall model structure of video-SALMONN S. The video encodings are first passed through the TTT$_{\text{HF}}$ layer (Section 3.2) followed by a similarity discarding procedure to keep fixed-size memory (Section 3.1). The fixed memory is then used as the input to the LLM, optionally using a prompt-dependent reading mechanism (Section 3.3). Audio tokens bypass the TTT$_{\text{HF}}$ layer.

et al., 2025) further improve efficiency via dynamic, non-uniform KV compression. Wang et al. (2025a) explores the concept of test-time training, but focuses on adaptation to local changes at test time that does not require long-term dependencies. In contrast, **video-SALMONN S** avoids hard token dropping/merging by using TTT to continually update the memory representations, mitigating information loss while meeting a fixed memory budget.

## 3 METHODOLOGY

The overall structure of video-SALMONN S is shown in Fig. 1. The input video is processed in streaming mode at a fixed frame rate, *e.g.* 1 FPS. Each video frame is first converted into video encodings using a visual encoder, followed by a pre-trained modality aligner that projects the representations to the text space. Video encodings are then fed into the TTT$_{\text{HF}}$ layer, which incorporates the full sequence history into the representations. We maintain a long-term memory of a fixed number of tokens, and the output of the TTT$_{\text{HF}}$ layer for the current frame provides new incoming tokens to be added to the memory. Thereafter, merging or downsampling can be performed to reduce the memory to the predefined size. This way, the memory requirement remains constant with the increasing input video length. The memory is then fed into the LLM to perform various tasks. Optionally, prompt-dependent memory reading can be applied in each Transformer block to select useful KV-Cache based on attention scores. The TTT$_{\text{HF}}$ layer and the low-rank adapter (LoRA) contain trainable parameters, and other parts of the model are frozen throughout training.

### 3.1 LONG-TERM MEMORY STRUCTURE

The long-term memory in video-SALMONN contains a TTT$_{\text{HF}}$ layer followed by a memory merging and discarding process to keep the total amount of memory unchanged. Specifically, given the encodings of a video frame denoted as $\mathbf{X}_t \in \mathbb{R}^{K \times d}$ where $K$ is the number of tokens, the incoming memory token to be added to the memory is derived as

$$\mathbf{Z}_t, \mathbf{W}_t = \text{TTT}_{\text{HF}}(\mathbf{X}_t, \mathbf{W}_{t-1}), \tag{1}$$

where $\mathbf{W}_{t-1}$ is the weight carrying history information and is updated to incorporate the information in $\mathbf{X}_t$. The detailed implementation is described in Section 3.2. The gating mechanism following Dalal et al. (2025) is used. When audio tokens are present, we bypass the TTT-layer since the number of tokens are usually much smaller, and directly append those tokens at the end of $\mathbf{Z}_t$.

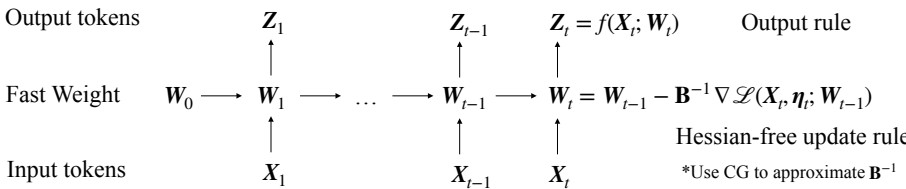

Figure 2: The overall workflow of the $\text{TTT}_{\text{HF}}$ layer. The layer works as an RNN model, which updates the current fast-weight $W_{t-1}$ of a MLP model for an incoming mini-batch of token $X_t$ to minimise a reconstruction loss. The Hessian-free method is used to construct the update. This MLP model is then used to generate an output token $Z_t$. The figure is adapted from Sun et al. (2024).

The memory tokens $\mathbf{Z}_t$ are then combined with previous memory tokens $\tilde{\mathbf{Z}}_{t-1} \in \mathbb{R}^{N \times d}$ where $N$ is the fixed number of memory tokens following a cosine similarity-based token discarding procedure (Song et al., 2024; Yang et al., 2025a). Specifically, let us denote $\mathbf{Z}'_t = \text{Concat}(\tilde{\mathbf{Z}}_{t-1}, \mathbf{Z}_t)$ along the sequence dimension, that is, $\mathbf{Z}'_t \in \mathbb{R}^{(N+K) \times d}$. We discard $K$ tokens that have the highest cosine similarities with their next tokens, *i.e.* $\cos(Z'_{t,n}, Z'_{t,n+1})$. The remaining $N$ tokens become the new memory tokens, $\tilde{\mathbf{Z}}_t$. Note that we particularly chose to discard rather than a merging operation as adopted in MovieChat (Song et al., 2024) to better exploit the long-term context representation ability of $\text{TTT}_{\text{HF}}$ layer and avoid over-smoothing in extremely long videos. Despite discarding tokens, information of those corresponding tokens is still retained by the $\text{TTT}_{\text{HF}}$ layer. When a prompt is used, the current memory tokens $\tilde{\mathbf{Z}}_t$ are sent to the LLM to either directly answer or use a prompt-dependent reading mechanism as described in Section 3.3 in detail.

## 3.2 TTT with Hessian-Free in video-SALMONN S

The overall structure of the TTT layer follows mainly the causal implementation in (Dalal et al., 2025), and the overall workflow is shown in Figure 2. The TTT layer consists of a multi-layer perceptron (MLP) model with fast weight $W$, and three learned projection matrices $\boldsymbol{\theta}_Q$, $\boldsymbol{\theta}_K$, and $\boldsymbol{\theta}_V$. The input to the $\text{TTT}_{\text{HF}}$ layer is a stream of mini-batches of consecutive tokens $X_1, X_2, \ldots X_t, \ldots$. For each incoming mini-batch $X_t$, the $\text{TTT}_{\text{HF}}$ layer does two things. Firstly, it updates its current fast weight $W_{t-1}$ by minimising the following reconstruction loss

$$\mathcal{L}(\boldsymbol{X}_t; \boldsymbol{W}_{t-1}) = \|f(\boldsymbol{\theta}_K \boldsymbol{X}_t; \boldsymbol{W}_{t-1}) - \boldsymbol{\theta}_V \boldsymbol{X}_t\|_2, \tag{2}$$

where $f(\cdot; \boldsymbol{W})$ represents the inference function of the MLP model, following the MLP instantiation with residual connection and layer norm (LN) operation by Dalal et al. (2025) as

$$f(\boldsymbol{x}; \boldsymbol{W}) = \boldsymbol{x} + \text{LN}(f_{\text{MLP}}(\boldsymbol{x}; \boldsymbol{W})). \tag{3}$$

The fast weight can be updated with a choice of optimizer, and the updated fast weight becomes $\boldsymbol{W}_t = \boldsymbol{W}_{t-1} + \Delta \boldsymbol{W}_t$. Secondly, the updated MLP model is used to produce the output token $\boldsymbol{Z}_t$,

$$\boldsymbol{Z}_t = f(\boldsymbol{\theta}_Q \boldsymbol{X}_t; \boldsymbol{W}_t). \tag{4}$$

The name of TTT comes from the fact that the fast weight $\boldsymbol{W}_t$ is updated for each incoming mini-batch at test time.

### 3.2.1 Approximate Second-Order Update in TTT

A key element of the TTT layer is the update $\Delta \boldsymbol{W}_t$ to its fast-weight. A standard choice is to generate the update with stochastic gradient descent (SGD) of the reconstruction loss as

$$\Delta \boldsymbol{W}_t^{\text{SGD}} = -\eta_t \nabla_{\boldsymbol{W}} \mathcal{L}(\boldsymbol{X}_t; \boldsymbol{W}_{t-1}). \tag{5}$$

It is known that approximate second-order (2nd) updates can improve learning efficiency greatly, which typically use the inverse of an approximate curvature matrix $\mathbf{B}$ to precondition the first-order gradient. The corresponding update rule is

$$\Delta \boldsymbol{W}_t^{\text{2nd}} = -\eta_t \mathbf{B}^{-1} \nabla_{\boldsymbol{W}} \mathcal{L}(\boldsymbol{X}_t; \boldsymbol{W}_{t-1}). \tag{6}$$

In the context of TTT, a range of popular second-order optimizers such as the K-FAC family (Martens & Grosse, 2015) is inapplicable because the explicit inverse operation applied on $\mathbf{B}$ can obstruct the gradient flow during the training of the TTT layer. One possible choice of second-order optimizer that does not involve explicit inverse operation is the Muon optimizer (Jordan et al., 2024), which is shown to be effective by Zhang et al. (2025d). In this paper, we explore a viable alternative second-order optimizer for TTT called Hessian-free (HF) by Martens (2010a).

The Hessian-free method avoids explicit inverse of $\mathbf{B}$ by using the linear CG algorithm to iteratively solve for the following equation

$$\mathbf{B}\Delta\boldsymbol{W}_t^{\mathrm{HF}} = -\eta_t\nabla_{\boldsymbol{W}}\mathcal{L}(\boldsymbol{X}_t; \boldsymbol{W}_{t-1}). \tag{7}$$

In each iteration, only the operation of the matrix vector product $\mathbf{B}(\cdot)$ is involved, which can be efficiently implemented for the MLP model of the TTT layer using the method proposed by Pearlmutter (1994). The details of the $\mathrm{TTT}_{\mathrm{HF}}$ update rule is shown in Algorithm 1, which is adapted from the CG algorithm in Wu et al. (2024).

### 3.3 PROMPT-DEPENDENT MEMORY READING

The video memory token sequence $\tilde{\mathbf{Z}}_t \in \mathcal{R}^{N \times d}$ after $\mathrm{TTT}_{\mathrm{HF}}$ already contains comprehensive information from the preceding video. However, when responding to a specific prompt $\mathbf{P}_t \in \mathcal{R}^{S \times d}$, it is usually unnecessary to utilise all memory tokens, which brings both performance issues and high computational cost. Therefore, we employ a prompt-dependent memory reading mechanism, following AdaReTaKe (Wang et al., 2025b), to select only the relevant part of the memory.

We aim to leverage the LLM to compress the number of KV pairs used for final decoding to an average of $M$ tokens per layer. Specifically, the tokens are first divided into chunks of length $m$, resulting in $\lceil N/m \rceil$ chunks, $\tilde{\mathbf{Z}}_{t,i}$, to reduce computational requirements. For each chunk at layer $l$, we concatenate the input of this chunk with prompt tokens forwarded to layer $l$, and perform the attention operation defined as

$$\mathbf{O}_i^{(l)} = \mathrm{Attention}\left(\mathrm{Concat}(\mathbf{X}_i^{(l)}; \mathbf{X}_{\mathrm{prompt}}^{(l)})\big|\mathbf{KV}_{1:i-1}^{(l)}\right), \tag{8}$$

where $\mathbf{X}_i^{(l)}$ are $\tilde{\mathbf{Z}}_{t,i}$ forwarded to layer $l$ and $\mathbf{X}_{\mathrm{prompt}}^{(l)}$ are $\mathbf{P}_t$ forwarded to layer $l$. $\mathbf{KV}_{1:i-1}$ are compressed KV cache carried over up to chunk $i$. We then compute the average attention score from the prompt to each position in chunk $i$ to reflect the importance of each KV pair as

$$\mathbf{a}_{t,i}^{(l)} = \sum_{s=1}^{S} \frac{1}{H} \sum_{h=1}^{H} \mathbf{A}_{t,i}^{(l)}[h, s], \tag{9}$$

where $\mathbf{A}_{t,i}^{(l)}[h, s]$ is the attention score from token $s$ to all vectors in $\mathbf{X}_i^{(l)}$ and $H$ is the number of attention heads. For all KV pairs across $L$ layers, $K' = m \times L \times M/N$ of them with the highest importance are reserved and appended to the KV list as

$$\mathrm{Indices} = \mathrm{ArgTopK}(\mathrm{Concat}(\mathbf{a}_{t,i}^{(1)}, \mathbf{a}_{t,i}^{(2)}, \ldots, \mathbf{a}_{t,i}^{(L)}), k = K') \tag{10}$$

$$\mathbf{KV}_{1:i}^{(l)} = \mathrm{Concat}(\mathbf{KV}_{1:i-1}^{(l)}, \mathbf{KV}_i^{(l)}[\mathrm{Indices}]), \tag{11}$$

where $\mathrm{ArgTopK}(\cdot, k = \cdot)$ selects the indices of the $k$ largest values. After $\lceil N/m \rceil$ iterations, all remained KV pairs, $\mathbf{KV}_t$, contains highly condensed content that is highly relevant to the current question. The LLM then generates a response $\hat{\mathbf{Y}}_t$ based on the prompt $\mathbf{P}_t$ and the extracted $\mathbf{KV}_t$:

$$\hat{\mathbf{Y}}_t = \arg\max_{\mathbf{Y}} P(\mathbf{Y}|\mathbf{KV}_t, \mathbf{P}_t). \tag{12}$$

The main benefit of prompt-dependent reading is to allow a much larger memory size to be used while still being feasible with the computational budget. This mimics the process of extracting task-specific information from long-term memory to working memory in the human brain (Jeneson & Squire, 2012; Chai et al., 2018). Since the memory size is fixed, it is still a streaming model that is suitable for videos of any lengths.

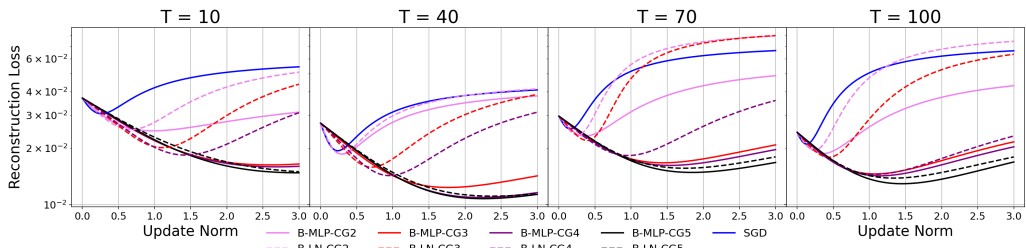

Figure 3: The reconstruction loss Eqn. (2) of single mini-batch sample *w.r.t.* update norm $\|\Delta \boldsymbol{W}_t\|_2$. Samples $\boldsymbol{X}_t$ (at index 10, 40, 70, 100) are from the same input sequence. Projection matrices $\boldsymbol{\theta}_K, \boldsymbol{\theta}_V$ are extracted from a trained TTT layer with standard SGD updates. The update generated by the SGD baseline and the HF method with curvature matrix $\mathbf{B}_{\text{MLP}}$ (B-MLP) and $\mathbf{B}_{\text{LN}}$ (B-LN), defined in Eqn. (13), are compared with CG iterations 2, 3, 4, and 5.

## 4 EXPERIMENTAL SETUP

### 4.1 DATA

We use LLaVA-Video-178k (Zhang et al., 2025e) as our video training set for fine-tuning the visual-only model. For audio-visual model, following Tang et al. (2025a), we first train the audio aligner using LibriSpeech 960-hour (Panayotov et al., 2015), CommonVoice (Ardila et al., 2020), WavCaps (Mei et al., 2024), and AudioCaps (Kim et al., 2019), with other parts of the LLM frozen. Then, FineVideo(Farré et al., 2024), CinePile Rawal et al. (2024) and about 13k videos with rich audio information from LLaVA-Video-178k are used to finetune the audio-visual model.

We evaluate video-SALMONN S on 4 different video question-answering benchmarks with video lengths ranging from a couple of minutes to several hours. **Video-MME** (Fu et al., 2024) is an audio-visual video QA benchmark featuring 3 different lengths of videos: short (less than 2 minutes), medium (around 15 minutes) and long (around 1 hour). **MLVU** (Zhou et al., 2025) mainly containing videos of around 10 minutes, requiring both holistic and detail-oriented understanding. **LVBench** and **VideoEvalPro** are two extremely long video QA benchmarks. LVBench (Wang et al., 2024b) contains videos of several hours that focusing on long-term memory and diverse core capabilities, and VideoEvalPro (Ma et al., 2025) also contains videos of several hours, and is designed to more realistically evaluate long video understanding without MCQ shortcuts, where the MCQ partition is used. Percentage accuracies are reported on all these benchmarks.

### 4.2 MODEL SPECIFICATIONS

Both visual-only and audio-visual models are built based on the Qwen2.5-VL 7B model (Bai et al., 2025). The video is processed at 1 FPS by default, and have an alternative 4 FPS for short videos less than 2 minutes (i.e. exclusively used for video-MME short partition). Following video-SALMONN-2 (Tang et al., 2025a), Whisper-Large-v3 encoder (Radford et al., 2023) is used to encode audio information and a window-level Q-Former with a window length of 0.5 seconds as the audio aligner. The LLM backbone is trained with LoRA with rank 128. We cap the memory tokens at 16k by default, and discuss in the experiments the influence of the memory size.

TTT$_{\text{HF}}$ processes the sequence with a mini-batch size of 1024. To determine CG update steps, the target curvature matrix $\mathbf{B}_{\text{MLP}}$ with 3 CG iteration is chosen based on empirical convergence analysis on single mini-batch shown in Figure 3, as it achieves the best balance between convergence and cost. Details of other design choices are provided in the Appendix A.1.

As points of comparison, we also use similarity merging alone without TTT$_{\text{HF}}$ as a baseline, where the merger module in Qwen2.5-VL is trained to encourage merging to happen in a better joint representation space. Note that this is equivalent to Li et al. (2025b) which is a highly effective technique for long or high-frame-rate video understanding. The merging process without trainable projection layer also resembles videoLLM-online Chen et al. (2024a). Moreover, we apply the standard Mamba-2 (Dao & Gu, 2024) implementation with default settings, and the standard LaCT (Zhang et al., 2025d) layer implementation as another two points of comparison to TTT$_{\text{HF}}$.

| Model | Video-MME (S/M/L) | MLVU | LVBench | VideoEvalPro |
|---|---|---|---|---|
| **Non-Streaming Models** | | | | |
| Qwen-2.5-VL 7B (Bai et al., 2025) | 65.5 (76.4/66.2/53.8) | 70.2 | 45.3 | 46.9 |
| Qwen-2.5-VL 7B + SFT | 67.5 (78.8/67.8/56.0) | 70.2 | 43.6 | 47.8 |
| Video-LLaMA-3 7B (Zhang et al., 2025a) | 66.2 (80.1/63.7/54.9) | 73.0 | 45.3 | - |
| LongVILA (Chen et al., 2024b) | 65.1 (72.9/64.9/57.4) | - | - | - |
| LLaVA-Video (Zhang et al., 2025e) | 63.3 ( - / - / - ) | - | - | - |
| v-SALMONN-2+ 7B SFT (Tang et al., 2025a) | 71.5 (79.9/72.6/62.1) | 69.8 | 44.1 | 49.7 |
| v-SALMONN-2+ 7B SFT + AdaReTaKe | 72.8 (**81.1**/71.7/65.1) | 72.6 | 50.2 | 54.9 |
| **Streaming Models** | | | | |
| Dispider (Qian et al., 2025) | 57.2 ( - / - / - ) | 61.7 | - | - |
| StreamMem (Yang et al., 2025a) | 62.4 ( - /62.4/52.3) | 65.9 | - | - |
| v-SALMONN S (ours, V, PI) | $68.3_{67.9}$ ($78.0_{77.0}$/68.3/58.3) | 73.2 | 47.1 | 51.1 |
| v-SALMONN S (ours, V, PD) | $69.3_{68.5}$ ($77.4_{75.2}$/69.6/60.8) | **73.2** | **52.8** | **55.8** |
| v-SALMONN S (ours, A+V, PI) | $71.0_{70.6}$ ($80.3_{79.1}$/71.3/61.3) | 70.8 | 46.8 | 48.5 |
| v-SALMONN S (ours, A+V, PD) | $\mathbf{74.2}_{73.8}$ ($80.3_{79.1}$/**74.4/67.8**) | 73.1 | 51.5 | 55.2 |

Table 1: video-SALMONN S compared to streaming and non-streaming baselines on Video-MME short/medium/long (S/M/L), MLVU, LVBench and VideoEvlaPro. Prompt-independent (PI) models directly feed 16k memory tokens to the LLM whereas prompt-dependent (PD) models uses 128k memory tokens and the memory reading. All baseline non-streaming models also use 16k visual tokens (145 frames), except for video-SALMONN 2+ SFT + AdaReTaKe which uses 2048 frames.

### 4.3 TRAINING AND INFERENCE CONFIGURATIONS

Visual only models are trained with a single fine-tuning stage. Parameters in the LoRA adapters and in $\text{TTT}_{\text{HF}}$ layers are updated. The visual-only model was trained for one epoch on the training data, and the audio-visual model was trained for three epochs, both with a learning rate of $2\times10^{-5}$. As the training videos are mainly less than 10 minutes, we use a 4 FPS sampling rate with a maximum of 1024 frames in total to achieve a better trade-off between parallelisation and sequence lengths. Training takes 32 hours and 48 hours on 32×H800 GPUs for the visual-only and audio-visual models, respectively. The batch size is 1 per GPU.

During inference, the prompt-dependent reading mechanism is applied, where much larger memory sizes are used (e.g. 64k or 128k). The average number of retained KV cache, $K'$, in Eqn. (10), is set to be 16k for the visual-only model and 24k for the audio-visual model. Inference of video-SALMONN S can be run on a single H800 GPU with 80GB memory, and the major time consumption is video loading. [1] Detailed runtime metrics are provided in Appendix D

## 5 RESULTS

### 5.1 MAIN RESULTS

We first present the main results of video-SALMONN S compared to a range of SOTA streaming and non-streaming baselines in Table 1. Specifically, for the visual-only non-streaming baselines, we include Qwen-2.5-VL 7B, which is the original pre-trained model that video-SALMONN S is adapted from. We also include the fine-tuned version of it with 16k tokens to eliminate the influence of instruction tuning data, as well as ensuring a fair comparison regarding the number of tokens. For audio-visual baselines, we include Video-LLaMA 3 and video-SALMONN 2+, fine-tuned on the same training data with 16k visual tokens as an SOTA baseline. Moreover, we use the same AdaReTaKe settings as the prompt-dependent reading mechanism for video-SALMONN 2+ as the strongest baseline across the table. We also compare two most recent streaming models, Dispider and StreamMem, on the test sets where numbers are reported. We follow previous work to demonstrate their long video understanding abilities under streaming mode on general benchmarks (Zeng et al., 2025; Yang et al., 2025a; Qian et al., 2025), since most streaming benchmarks are focused on short videos and real-time understanding (Lin et al., 2024b; Li et al., 2025a; Yang et al.,

---
[1]Our data, model checkpoint and code will be made open-source.

| Configuration | Video-MME (S/M/L) | MLVU | LVBench |
|---|---|---|---|
| Similarity Merging | 67.1 (77.6/67.9/55.9) | 72.5 | 45.7 |
| K-means Clustering (Zhou et al., 2024) | 66.8 (77.9/67.5/54.9) | 70.5 | 43.6 |
| Similarity Discarding | 67.2 (78.0/67.8/56.0) | 71.9 | 45.6 |
| + Mamba-2 (Dao & Gu, 2024) | 67.9 (77.0/69.9/57.2) | 72.9 | 46.7 |
| + $\text{TTT}_{\text{SGD}}$ | 67.7 (76.6/68.6/58.1) | 73.1 | 46.9 |
| + $\text{TTT}_{\text{HF}}$ (video-SALMONN $S_{\text{PI}}$) | 67.9 (77.0/68.3/58.3) | 73.2 | 47.1 |
| + PD | 67.8 (75.9/68.6/58.6) | 72.6 | 51.6 |
| + $\text{TTT}_{\text{SGD}}$ + PD | 68.2 (77.3/68.7/58.7) | 72.8 | 52.4 |
| + $\text{TTT}_{\text{HF}}$ + PD (video-SALMONN $S_{\text{PD}}$) | 68.5 (75.2/69.6/60.8) | **73.2** | **52.8** |

Table 2: Ablation studies on different components in video-SALMONN S for visual-only models. $\text{TTT}_{\text{SGD}}$ refers to the standard TTT implementation in (Sun et al., 2024).

2025b), where long-term memory is less important. For completeness, we also provide results on a dedicated online video understanding benchmark in Appendix B, where video-SALMONN S also achieved competitive performance and the proposed memory mechanism showed improvements when there is long-term memory needed.

As a result, **video-SALMONN S achieves the best performance** across the table, even surpassing the offline counterpart. This mainly benefits from the completeness of information, which is more important when the video gets longer. Specifically, while both use 16k tokens, the video-SALMONN S prompt-independent (PI) visual-only model outperforms Qwen-2.5-VL 7B finetuned on the same data but with a fixed number of frames, and the video-SALMONN $S_{\text{PI}}$ audio-visual model outperforms video-SALMONN 2+. Notably, even with AdaReTaKe to scale up the number of frames to 2048 for video-SALMONN 2+, prompt-dependent (PD) video-SALMONN S still achieves clearly better performance, which can be attributed to the $\text{TTT}_{\text{HF}}$ layer. In summary, video-SALMONN $S_{\text{PD}}$ with audio-visual inputs achieved state-of-the-art performance across both streaming and non-streaming models, especially on video-MME benchmark with **74.2**% overall accuracy and **67.8**% accuracy on the long partition.

**Performance boosts are mainly in long videos**: For prompt-independent models, compared to Qwen-2.5 VL SFT, video-SALMONN S yields a slightly worse performance on video-MME short (78.0 vs. 78.8) and on-par performance on video-MME medium (68.3 vs. 67.8), and obviously better performance on the other three benchmarks. Similar phenomenon is observed for audio-visual models. This is mainly due to the sparsity of the frames for long videos. For prompt-dependent models, as more frames can be involved for the offline model hence reducing information loss, the gap becomes slightly smaller but still consistent across all four benchmarks. In terms of GPU memory, our design maintains a fixed-size memory representation and processes frames in a streaming manner. As a result, the GPU memory required at test time does not scale with the number of frames, even when evaluating hour-long videos.

## 5.2 Ablation Studies

**Investigating influence of different design choices in video-SALMONN S**: We perform ablations studies on different components and design choices first, using the visual-only model to eliminate the effect of audio as it bypasses $\text{TTT}_{\text{HF}}$. As shown in Table 2, we start with the basic similarity merging baseline resembling MovieChat (Song et al., 2024). There is no statistical significant difference between the results using token merging or discarding if the merger module in Qwen2.5-VL is trained. Adding $\text{TTT}_{\text{SGD}}$ achieved clear performance improvements as TTT provides a better memory mechanism than token merging, which is also surperior to a Mamba-2 layer. As before, salient improvements of TTT modules are found mainly on long videos, with $\text{TTT}_{\text{HF}}$ achieving consistently better performance compared to any other prompt-independent memory models.

When prompt-dependent reading is applied, clear improvements on long videos are observed since a larger memory size of 128k tokens is used. With a similar amount of performance gains brought about by PD on all models, the advantage of $\text{TTT}_{\text{HF}}$ persists, achieving the best performance across the table, which validates our design choice.

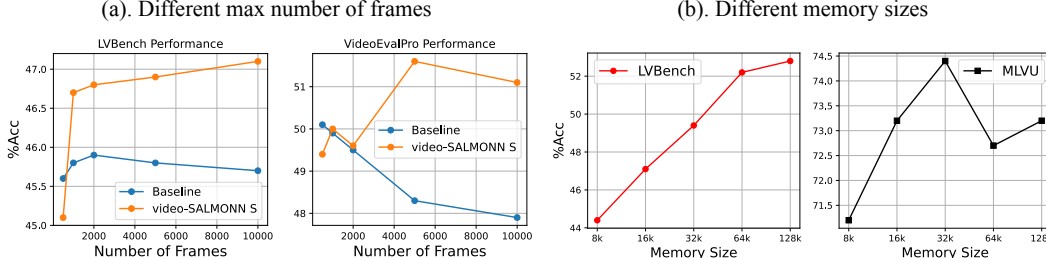

Figure 4: Ablation studies on (a). The influence of the maximum number of frames on two extremely long video benchmarks (Left: LVBench, Right: VideoEvalPro), and (b). The influence of memory size on video-SALMONN S. When memory size exceeds 32k, prompt-dependent reading is used. Baseline refers to similarity merging without prompt-dependent reading.

| Model | Video-MME (S/M/L) | MLVU | LVBench | VidEvalPro |
|---|---|---|---|---|
| TTT$_{SGD}$ | 67.7 (76.6/68.6/58.1) | 73.1 | 46.9 | 50.8 |
| TTT$_{Muon}$ | 67.6 (76.8/68.8/57.1) | 72.8 | 46.2 | 49.9 |
| LaCT (Zhang et al., 2025d) | 67.8 (77.4/69.1/57.0) | 72.7 | 46.0 | 50.4 |
| TTT$_{HF}$ | **67.9** (77.0/68.3/58.3) | **73.2** | **47.1** | **51.1** |

Table 3: Comparing TTT$_{HF}$ to other TTT optimisation methods on visual-only models across all benchmarks. We implement Muon ourselves while keeping other configurations the same as TTT$_{HF}$, and we also compare to the default LaCT configuration, which differs in general from ours.

**video-SALMONN S benefits more from a higher number of frames**: We then show the influence when the maximum number of frames is limited in Figure 4 (a) for the two extremely long video benchmarks. That is, if the number of frames at 1 FPS for a video exceeds the maximum frame, we evenly down sample frames to satisfy the limit. As shown, the baseline reaches the maximum very early at 1000 frames on LVBench, and only degrades when the number of frames increases for VideoEvalPro. In contrast, the video SALMONN S exhibits clear upward trends with more frames on both benchmarks, showcasing a more effective use of history information.

**Investigation into memory sizes**: The memory size is determined by the number of tokens, which by default is 16k for video-SALMONN S. Smaller number of tokens, *e.g.* 8k or 6k, are often used in previous literature. To allow more direct comparison to other systems, we also provide the results with 8k, 32k and larger memory sizes in Table 6 in Appendix C in detail, and show the variation of performance against the memory size on MLVU (medium-length videos) and LVBench (long videos) in Figure 4 (b) for the visual-only video-SALMONN S model.

The visual-only model can handle up to 32k memory tokens on a single GPU using FlashAttention 3 (Shah et al., 2024) under our setup, and a linear upward trend is observed when the memory size increases from 8k to 32k for MLVU, and 8k to 64k for LVBench, directly showing the benefit of using a memory larger than 8k under a single-GPU budget. Meanwhile, the improvements beyond 64k memory are marginal since **(i)** the length of videos in current benchmarks does not require such large memory, and **(ii)** the post-training prompt-dependent memory reading is less effective than trained prompt-independent memory. While 32k memory performs better, the audio-visual model exceeds the memory limit for single GPU and hence 16k memory was adopted for the main result. We will provide checkpoints for the visual-only model with 8k and 32k memory tokens as well.

**TTT$_{HF}$ outperforms TTT$_{Muon}$ as a memory module**: We conduct ablations studies to investigate the influence of optimizer choices in TTT-layer, as shown in Table 3. We observe that TTT$_{HF}$ > TTT$_{SGD}$ > TTT$_{Muon}$/LaCT (both of which use the Muon optimizer). The implementation details of TTT$_{Muon}$ are provided in Appendix A.2.1. It is surprising that TTT$_{Muon}$, which is commonly considered a better option for TTT-layer (Zhang et al., 2025d; Behrouz et al., 2025), performs worse than TTT$_{SGD}$. An analysis on the statistics during TTT inference is provided in Appendix A.2.2. It shows that the performance gap might be caused by the nature of the Muon update. As is shown in (Wu et al., 2024), second-order optimizers achieve better convergence of the training loss by limiting the impact of their parameter update on the model output.

For Muon (see Figure 5 in Appendix A.2.2), it achieves the lowest final reconstruction loss but is unable to effectively modify the output of the TTT-layer. In other words, updates of the $TTT_{Muon}$ pass less information to the downstream LLM, likely making the downstream training more difficult to learn. In comparison, $TTT_{HF}$ achieves a better balance between convergence and information flow (than both $TTT_{Muon}$ and $TTT_{SGD}$), thus delivering the better final results across benchmarks.

## 6 CONCLUSION

This paper proposes video-SALMONN S, the first audio-visual LLM supporting streaming under-standing of >**3-hour** videos at **1 FPS** and **360p resolution** under a fixed memory budget. video-SALMONN S features a similarity discarding long-term memory mechanism powered by a novel $TTT_{HF}$ layer design dedicated to long-term context modeling, and a prompt-dependent memory reading mechanism. As a result, video-SALMONN S handles videos of several hours with over 1M tokens, and notably, achieves SOTA performance of 7B/8B models on video-MME, surpassing both streaming and non-streaming systems.

## 7 ETHICS STATEMENT

Our work introduces a frontier audio-visual large language model with strong video understanding capabilities. The model inherits potential risks from its backbone models, which are clearly refer-enced in the paper. Potential harms include biased outputs, misuse for disinformation, and safety vulnerabilities. To mitigate these, we conducted bias and robustness evaluations, applied content fil-tering, and chose a controlled release strategy. Training incurred notable computational cost, which we report in the paper, estimated the environmental impact and adopted the most efficient tech-niques where we can. Data sources were curated to exclude sensitive information and comply with licenses. While the model enables positive applications in science, accessibility, and collaboration, we recognise open ethical challenges and encourage community oversight.

## 8 REPRODUCIBILITY STATEMENT

We provide detailed descriptions of the model architecture, training objectives, datasets, and pro-cessing steps in Section 4.1 and appendices. Code for data preprocessing, training, and evaluation, along with configuration files and random seeds, will be released to facilitate exact reproduction. Model checkpoints will also be released to reproduce the results listed in our paper.

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

# A  DESIGN CHOICES AND FURTHER ANALYSIS OF TTT$_{\text{HF}}$

## A.1  DISCUSSIONS ON DESIGN CHOICES

The key design choices of TTT$_{\text{HF}}$ are explained and discussed in this section.

**Choice of curvature matrix B**  A crucial part of HF optimizer is the choice of the curvature matrix **B** in equation 7. As is mentioned in equation 3, the complete inference function of the TTT-layer is

$$f(\boldsymbol{x}; \boldsymbol{W}) = \boldsymbol{x} + \text{LN}(f_{\text{MLP}}(\boldsymbol{x}; \boldsymbol{W})),$$

where $\text{LN}(\cdot)$ is the layer norm operation. There are two potential choices of curvature matrices:

$$\mathbf{B}_{\text{MLP}} = (\nabla_{\boldsymbol{W}} \boldsymbol{z}_{\text{MLP}})^{\top} \nabla_{\boldsymbol{W}} \boldsymbol{z}_{\text{MLP}} \tag{13a}$$

$$\mathbf{B}_{\text{LN}} = (\nabla_{\boldsymbol{W}} \boldsymbol{z}_{\text{LN}})^{\top} \nabla_{\boldsymbol{W}} \boldsymbol{z}_{\text{LN}}, \tag{13b}$$

---

**Algorithm 1** The Fast-Weight Update Rule in $\text{TTT}_{\text{HF}}$ (Adapted from Wu et al. (2024))

---

**Input:** The maximum CG execution iterations: $M_{\text{CG}}$ (chosen to be 3 in this project)
The gradient of the $\eta$-scaled TTT reconstruction loss at time step $t$: $\nabla_{\boldsymbol{W}} \mathcal{L}(\boldsymbol{X}_t, \boldsymbol{\eta}_t; \boldsymbol{W}_{t-1})$
The matrix-vector product function for the Gauss-Newton matrix of the TTT-MLP model: $\mathbf{B}(\cdot)$
**Execute:**
$\boldsymbol{r}_0 \leftarrow \nabla_{\boldsymbol{W}} \mathcal{L}(\boldsymbol{X}_t, \boldsymbol{\eta}_t; \boldsymbol{W}_{t-1}), \boldsymbol{v}_0 \leftarrow \boldsymbol{r}_0, m \leftarrow 0$
**while** $m < M_{\text{CG}}$ **do**
    $\|\boldsymbol{r}_m\|^2 = \boldsymbol{r}_m^\top \boldsymbol{r}_m$
    $\alpha_m \leftarrow \|\boldsymbol{r}_m\|^2 / (\boldsymbol{v}_m^\top \mathbf{B} \boldsymbol{v}_m)$
    $\boldsymbol{x}_{m+1} \leftarrow \boldsymbol{x}_m + \alpha_m \boldsymbol{v}_m$
    $\boldsymbol{r}_{m+1} \leftarrow \boldsymbol{r}_m - \alpha_m \mathbf{B} \boldsymbol{v}_m$
    $\|\boldsymbol{r}_{m+1}\|^2 = \boldsymbol{r}_{m+1}^\top \boldsymbol{r}_{m+1}$
    $\beta_{m+1} \leftarrow \|\boldsymbol{r}_{m+1}\|^2 / \|\boldsymbol{r}_m\|^2$
    $\boldsymbol{v}_{m+1} \leftarrow \boldsymbol{r}_{m+1} + \beta_{m+1} \boldsymbol{v}_m$
    $m \leftarrow m + 1$
    **if** $\gamma(\boldsymbol{x}_{m+1}) > \gamma(\boldsymbol{x}_m)$ **then**
        $\boldsymbol{x}_{M_{\text{CG}}} = \boldsymbol{x}_m$
        **break**
    **end if**
**end while**
$\boldsymbol{W}_{t+1} \leftarrow \boldsymbol{W}_t - \boldsymbol{x}_{M_{\text{CG}}}$
**Return**: $\boldsymbol{W}_{t+1}$

---

where $\boldsymbol{z}_{\text{MLP}} = f_{\text{MLP}}(\boldsymbol{x}; \boldsymbol{W})$ and $\boldsymbol{z}_{\text{LN}} = \text{LN}(f_{\text{MLP}}(\boldsymbol{x}; \boldsymbol{W}))$. Note that the additional inference and backward operation of $\mathbf{B}_{\text{LN}}$ makes the matrix-vector product operation of $\mathbf{B}_{\text{LN}}(\cdot)$ 14.0% more computationally expensive than $\mathbf{B}_{\text{MLP}}(\cdot)$ (tested on Nvidia A5000 GPU).

To compare the quality of updates generated with these two choices of curvature matrices (along with the choice of number of CG iterations), we conduct a single batch empirical convergence analysis. The experiment observes the loss change along different update directions w.r.t. the update norm. As shown in Figure 3, when a small CG iteration number is used ($\leq 3$), $\mathbf{B}_{\text{MLP}}$ achieves better convergence (a consistently lower lowest achievable loss). Additionally, considering the additional computational cost of $\mathbf{B}_{\text{LN}}(\cdot)$, $\mathbf{B}_{\text{MLP}}$ is a much more cost-effective choice.

**Choice of CG iterations**    The CG algorithm is the most expensive part of $\text{TTT}_{\text{HF}}$, and the iteration number should be carefully chosen. Each CG iteration is equivalent to the computation cost of two forward passes of the $f_{\text{MLP}}(\cdot)$ function in equation 3. The optimal choice should be the smallest iteration number that leads to the most optimisation performance gain. As is shown in Figure 3, 3 CG iterations demonstrates the most convergence improvement (from 2 CG iterations), while further increasing the CG iterations shows diminishing gain.

**Incorporation of token-dependent learning rate**    Following the implementation in Dalal et al. (2025), for each incoming mini-batch, a token-dependent learning rate $\eta_t^i = h(\boldsymbol{X}_i)$ is assigned to each individual token. For the TTT-layer with SGD update, the accumulated update naturally becomes $\Delta \boldsymbol{W}_t^{\text{SGD}} = -\sum_i \eta_t^i \nabla_{\boldsymbol{W}} \mathcal{L}(\boldsymbol{X}_t^i; \boldsymbol{W}_{t-1})$. However, it is non-trivial as in how to incorporate this token-dependent learning rate in $\text{TTT}_{\text{HF}}$. In this paper, the token-dependent learning rate $\eta_t^i$ is re-interpreted as an importance measure for each token, and the TTT reconstruction loss is rewritten as an adaptive loss function with an adaptive per-token weighting $\mathcal{L}(\boldsymbol{X}_t, \boldsymbol{\eta}_t; \boldsymbol{W}_{t-1}) = \sum_i \eta_t^i \mathcal{L}(\boldsymbol{X}_t^i; \boldsymbol{W}_{t-1})$. This formulation alleviates the need to reweight the per-sample component of the curvature matrix of the TTT-MLP model when doing CG iterations, meanwhile maintaining consistency with standard TTT update methods such as SGD and Muon.

**Early stopping of CG iterations**    During update generation process of the $\text{TTT}_{\text{HF}}$ layer, an independent CG algorithm is run for every MLP model of each head (and sample) (see Algorithm 1, which describe one independent CG iterations). The CG iterations sometimes need to be terminated, or divergence would happen (e.g. when inversion is completed). This is determined mainly by the rank of the curvature matrix $\mathbf{B}$ of each MLP model and computation precision, which may

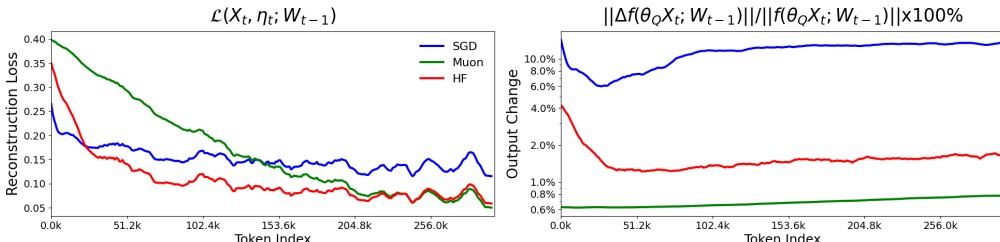

Figure 5: The TTT statistics of an one hour video during inference is shown in this figure. The sequence is run by the checkpoint trained with TTT$_{\text{Muon}}$, with a learnt per-time-step update norm of 0.1386. To enable fair comparison, the sequence is re-run through the TTT layer with the optimizer replaced with HF and SGD, meanwhile enforcing the same per-time-step update norm. On the left depicts the reconstruction loss $\mathcal{L}(\boldsymbol{X}_t, \boldsymbol{\eta}_t; \boldsymbol{W}_{t-1})$ and on the right depicts the TTT relative output change $\frac{\|\Delta f(\boldsymbol{\theta}_Q \boldsymbol{X}_t; \boldsymbol{W}_{t-1})\|}{\|f(\boldsymbol{\theta}_Q \boldsymbol{X}_t; \boldsymbol{W}_{t-1})\|}$. It can be observed that, although TTT$_{\text{Muon}}$ achieves the lowest reconstruction loss (by a small margin), the change incurred by each update to the output of the TTT-layer is significantly lower. This indicates the possibility that less information is incorporated into the output of the TTT$_{\text{Muon}}$ layer, which makes the learning task of downstream LLM more difficult.

vary greatly. It is therefore important to automatically determine if a CG iteration should end. In this paper, the $\gamma$ indicator proposed in Wu et al. (2024) is used, which is shown to monotonically decrease for the CG update $\boldsymbol{x}_m$ at the end of every CG iteration, until divergence. The monotonicity of $\gamma(\boldsymbol{x}_m)$ is monitored for every CG iteration, and an iteration is terminated once the monotonicity breaks.

### A.2 COMPARING TTT$_{\text{MUON}}$, TTT$_{\text{HF}}$ AND TTT$_{\text{SGD}}$

A plausible explanation is provided in this section for the performance result in Table 3 (TTT$_{\text{HF}}$ > TTT$_{\text{SGD}}$ > TTT$_{\text{Muon}}$).

#### A.2.1 DETAILS OF TTT$_{\text{MUON}}$

The Muon optimizer (Jordan et al., 2024) is an emerging optimizer which is designed specifically for 2D weight matrices. It uses Newton-Schulz (NS) iterations to orthogonalise a target weight matrix. The Muon optimizer can be applied to each weight matrices of the MLP model in the TTT-layer, and the update can be generated as follows

$$\Delta \boldsymbol{W}_t^{\text{Muon}} = -\eta^{\text{Muon}} \text{NS}(\Delta \boldsymbol{W}_t^{\text{SGD}}), \tag{14}$$

where $\Delta \boldsymbol{W}_t^{\text{SGD}}$ represents the SGD update defined in equation 5. We follow the implementation of the NS iterations in Zhang et al. (2025d). Because the NS iteration removes all norm information of the input update matrix $\Delta \boldsymbol{W}_t^{\text{SGD}}$, a learnable parameter $\eta^{\text{Muon}}$ is added in our TTT$_{\text{Muon}}$ implementation to enable variable update norm during TTT. The final result of model trained with TTT$_{\text{Muon}}$ is shown in Table 3. It is surprising that TTT$_{\text{Muon}}$, along with the LaCT baseline (which also uses Muon in its TTT-layer), underperforms both our TTT$_{\text{HF}}$ and TTT$_{\text{SGD}}$ implementations.

#### A.2.2 UNDERSTANDING THE PERFORMANCE DIFFERENCE AMONG TTT$_{\text{MUON}}$, TTT$_{\text{SGD}}$ AND TTT$_{\text{HF}}$

To understand the performance gap between TTT$_{\text{Muon}}$ and our TTT$_{\text{HF}}$ and TTT$_{\text{SGD}}$, the per-update statistics during the inference of an one-hour video input is analysed (see Figure 5). Note that the right subplot depicts the change to TTT-layer output incurred by the update at each time step, i.e.

$$\frac{\|\Delta f(\boldsymbol{\theta}_Q \boldsymbol{X}_t; \boldsymbol{W}_{t-1})\|}{\|f(\boldsymbol{\theta}_Q \boldsymbol{X}_t; \boldsymbol{W}_{t-1})\|}, \tag{15}$$

where $\Delta f(\boldsymbol{\theta}_Q \boldsymbol{X}_t; \boldsymbol{W}_{t-1}) = f(\boldsymbol{\theta}_Q \boldsymbol{X}_t; \boldsymbol{W}_t) - f(\boldsymbol{\theta}_Q \boldsymbol{X}_t; \boldsymbol{W}_{t-1})$. This metric can be viewed as a rough estimation of the amount of new information injected into the output of the TTT-layer by the TTT update at each time step. By looking at the final reconstruction loss, it can be seen that,

convergence-wise, Muon $\geq$ HF $>$ SGD. However, Muon updates inject significantly less information to the output of the TTT-layer than both HF and SGD. This is an expected behavior, as all second-order update improves convergence by reducing its affect on model output behaviour (Wu et al., 2024). However, this could be the main pitfall for Muon, which likely makes the adaptation of downstream LLM to the TTT outputs much more difficult. On the other hand, HF achieves a better balance between convergence on reconstruction loss and information injection to TTT-layer output (better convergence on reconstruction loss than SGD, and higher output change than Muon), thus achieving the best overall performance (see Table 2 and 3).

## B  ADDITIONAL RESULTS ON ONLINE VIDEO BENCHMARK

We provide the performance of video-SALMONN S on an online video understanding benchmark, OVO-Bench (Li et al., 2025a), as shown in Table 5. Note that video-SALMONN S has never trained on online video understanding tasks or prompts, hence significant domain mismatch exists. Meanwhile, the videos in the benchmark is not long enough to completely reflect the advantage of our proposed mechanism. Nevertheless, video-SALMONN S still achieves the best performance across the table, compared to both offline and online systems.

| Model | Realtime | Backward | Forward | Overall |
|---|---|---|---|---|
| Non-Streaming Models | | | | |
| Qwen2-VL-7B Wang et al. (2024a) | 56.0 | 46.5 | 48.7 | 50.4 |
| LLaVA-Video-7B Zhang et al. (2025e) | 63.5 | 40.4 | **54.8** | 52.9 |
| Qwen2.5-VL-7B SFT | 63.4 | **48.0** | 47.2 | 52.9 |
| Streaming Models | | | | |
| Flash-VStream Zhang et al. (2025b) | 28.4 | 27.4 | 45.1 | 33.6 |
| Dispider Qian et al. (2025) | 54.6 | 36.1 | 34.7 | 41.8 |
| video-SALMONN S | **63.8** | 43.4 | 53.6 | **53.6** |

Table 4: Performance comparison on OVO-Bench for online video understanding.

We then provide the performance of video-SALMONN S on StreamingBench (Lin et al., 2024b).

| Model | Real-Time | Omni-Source | Contextual | Overall |
|---|---|---|---|---|
| Dispider Qian et al. (2025) | 54.6 | 36.1 | 34.7 | 41.8 |
| ViSpeak S2 (Fu et al., 2025) | 74.4 | **53.5** | **39.6** | 62.0 |
| Qwen2.5-VL (V) | 75.4 | 38.7 | 32.1 | 57.6 |
| Qwen2.5-VL Merging (V) | 75.1 | 39.1 | 33.8 | 57.9 |
| video-SALMONN S (V, PI) | 75.7 | 40.7 | 35.7 | 58.7 |
| video-SALMONN S (A+V, PI) | **76.0** | 53.0 | 36.1 | **62.1** |

Table 5: Performance comparison on StreamingBench for online video understanding.

## C  DETAILED RESULTS ON THE INFLUENCE OF MEMORY SIZES

We provide the complete results for the performance variation against memory size for video-SALMONN S in Table 6.

## D  RUNTIME METRICS AND ANLYSIS

For a TTT-layer with input dimension $d$, intermediate dimension $m$, total sequence length $T$, mini-batch size $B$ and minibatch size $B$, all three methods share the same core operations: For all the methods: TTTSGD, TTTMuon, TTTHF, they all require the following common operations:

- Forward operation with pre-updated fast weight ($2 \times Tdm$)

| Model (Memory Size) | VMME (S/M/L) | MLVU | LVBench |
|---|---|---|---|
| Baseline (8k) | 66.3 (76.8/66.9/55.3) | 70.6 | 41.6 |
| video-SALMONN S (8k) | 66.7 (76.9/66.2/57.1) | 71.2 | 44.4 |
| Baseline (16k) | 67.1 (77.6/67.9/55.9) | 72.5 | 45.7 |
| video-SALMONN S (16k) | 67.9 (77.0/68.3/58.3) | 73.2 | 47.1 |
| video-SALMONN S (32k) | **68.9** (76.8/70.9/59.0) | **74.4** | 49.4 |
| video-SALMONN S (64k, PD) | 68.6 (75.2/70.2/60.4) | 72.7 | 52.2 |
| video-SALMONN S (128k, PD) | 68.5 (75.2/69.6/60.8) | 73.2 | **52.8** |

Table 6: Influence of varying memory size for video-SALMONN S, where baseline refers to similarity merging and 8k/16k token versions are provided as those are commonly seen in literatures.

- Backward operation to compute the gradient ($4 \times Tdm$)
- Meta-optimiser-specific update generation cost (different across methods)
- Final forward operation for inference ($2 \times Tdm$)

This yields a constant base cost $8 \times Tdm$ for all variants. Extra costs caused by different optimisers are:

- SGD: no extra FLOPs for generating update.
- Muon: 5 NS iterations, each with 3 parameters, which in total costs $T \times (10 \times dm + 20 \times d^2) \times (d/B)$
- TTT-HF: 3 matrix vector products with the curvature matrix, which in total costs $18 \times Tdm$

Overall, TTT-HF has competitive computation complexity as compared to TTT-Muon (depending on the ratio of $d/B$, HF becomes more efficient than Muon for a smaller $B$). TTT-HF costs around 2.25x more when compared to TTT-SGD, but since the TTT layer is a single small MLP applied only to input tokens, the added FLOPs are negligible relative to the LLM backbone's inference cost.

We provide runtime metrics, including TFLOPS, inference time taken per frame during prefill, and first token latency, in Table 7

| Configuration | TFLOPS | Inference time per frame (s) | First token latency (s) |
|---|---|---|---|
| Qwen2.5 VL + Merge | 2.76 | 0.00398 | 0.799 |
| + TTT-SGD | 2.76+0.000206 | 0.00409 | 0.802 |
| + TTT-HF (3 steps) | 2.76+0.000668 | 0.00436 | 0.805 |

Table 7: Runtime metrics for video-SALMONN S measured on a single H800 GPU.

Key observations:

- TFLOPs overhead is negligible compared to the entire LLM because it is just a single MLP.
- Inference time per frame overhead is negligible compared to the first token latency.
- TTT-HF does not influence first token latency because it updates memory whenever new frames come. So when a user prompt comes, the visual memory is always ready to be used by the LLM.

As a result, with TTT-HF, for 1-hour video at 1FPS and 360p, we can still process within 20 seconds on average. We added this table to the revised paper.

