# OpenReview forum: "video-SALMONN S: Streaming Audio-Visual LLMs Beyond Length Limits via Memory"
_ICLR.cc/2026/Conference — Submitted to ICLR 2026_

### Official Review · Reviewer_U5i5 · 2025-10-29

**Soundness:** 2
**Presentation:** 2
**Contribution:** 2
**Rating:** 4
**Confidence:** 3

**Summary:**

This paper proposes a streaming-capable video-LLM architecture that processes extremely long videos via a test-time training (TTT) layer and a prompt-dependent memory-reading module. The TTT layer stores the salient history of previously seen inputs (similar to RNNs) and is optimized specifically with a Hessian-free second-order function. The outputted tokens are then further pruned using a prompt-dependent memory reading module. Extensive experiments, ablations, and discussions are provided to motivate certain choices made with their method, and quantitative results are shown on three long-video benchmarks. Specifically, they show their method can stream extremely long videos with stable accuracy and memory cost.

**Strengths:**

1. **Thorough discussion & ablations** - The authors provide detailed analysis of their design decisions: e.g., optimizer choice (SGD vs Muon vs. HF), optimal number of CG iterations, comparison with alternative methods of token-merging such as Mamba, etc. This supports the authors' specific design choices for each of their components.
2. **Strong Video-MME performance** - The reported performance of video-SALMONN S on Video-MME appears to SOTA for 7B-class models.
3. **Value of streaming extremely long videos** - The streaming capability of video-SALMONN S for very long-context is a strong practical contribution.

**Weaknesses:**

1. **Heavy reliance on prior work/limited novelty** - The authors transparently note that core modules (TTT layer, memory reader) are mostly borrowed from [1] and AdaReTaKe. While some of their adaptations are non-trivial, the novelty is still somewhat limited. The memory reader closely mirrors AdaReTaKe’s idea of prompt-guided KV selection, but adapts it for streaming long videos and modifies the budgeting strategy (one global Top-K rather than layer/time budgets). This adaptation is reasonable but modest.  Further, the TTT layer is kept exactly the same but simply trained with a different optimizer (see below).

2. **Missing/insignificant quantitative comparisons** - While I understand the proposed method is specifically tailored for streaming long videos, the authors do not provide a thorough comparison against non-streaming / long video understanding VLMs. For example, [2, 3] outperform almost every baseline (including video-SALMONN S) presented in this paper on LVBench, has competitive performance on MLVU, and is not included in the paper. Why are the improvements on LVBench negligible? Moreover, v-SALMONN-2 already appears to have competitive performance with v-SALMONN S across all datasets besides Video-MME, calling into question how much of an improvement the streaming aspect provides in long video understanding. This could potentially stem from long video datasets being unfit for truly multi-hour-long video streaming benchmarking.
On a similar note, some presented results show that improvements are insignificant. For example, Table 3 shows that the optimizer used in the TTT layer doesn't seem to really have any significant impact on the results (SGD vs. HF), yet is the only change the authors propose for the TTT layer and is discussed in detail in Section 3.2.

[1] Dalal, K., Koceja, D., Xu, J., Zhao, Y., Han, S., Cheung, K. C., ... & Wang, X. (2025). One-minute video generation with test-time training. In Proceedings of the Computer Vision and Pattern Recognition Conference (pp. 17702-17711).

[2] Chen, Y., Xue, F., Li, D., Hu, Q., Zhu, L., Li, X., ... & Han, S. LongVILA: Scaling Long-Context Visual Language Models for Long Videos. In The Thirteenth International Conference on Learning Representations.

[3] Zhang, Y., Wu, J., Li, W., Li, B., Ma, Z., Liu, Z., & Li, C. (2024). Video Instruction Tuning With Synthetic Data. In Transactions on Machine Learning Research

**Questions:**

1. Can the authors further motivate why their main contributions to the TTT layer and memory reader are significant? It appears the changes to TTT are trivial, and while the memory reader further equips v-SALMONN S to stream long contexts, it does not seem to yield large improvements on long video benchmarks.
2. Why are certain non-streaming model comparisons (see Weaknesses) left out? Either include all of them as comparisons, or discuss/exhibit why they are unfit for long-video understanding and how v-SALMONN S outperforms them.

I would be happy to increase my score if these questions (and those discussed in Weaknesses) are addressed.

---

> ### Author Response · Authors · 2025-11-19
> **Response to Reviewer U5i5**
>
> We thank the reviewer for the positive comments and would like to clarify and address each concern as follows:
>
> > Weakness 1: Heavy reliance on prior work/limited novelty - The authors transparently note that core modules (TTT layer, memory reader) are mostly borrowed from [1] and AdaReTaKe. While some of their adaptations are non-trivial, the novelty is still somewhat limited. The memory reader closely mirrors AdaReTaKe’s idea of prompt-guided KV selection, but adapts it for streaming long videos and modifies the budgeting strategy (one global Top-K rather than layer/time budgets). This adaptation is reasonable but modest. Further, the TTT layer is kept exactly the same but simply trained with a different optimizer.
>
> > Question 1: Can the authors further motivate why their main contributions to the TTT layer and memory reader are significant? It appears the changes to TTT are trivial, and while the memory reader further equips v-SALMONN S to stream long contexts, it does not seem to yield large improvements on long video benchmarks.
>
> Thank you for the critique. We clarify why our contributions to both the TTT layer and the memory reader are substantial rather than trivial.
> 1. __Our modification to the TTT layer is structurally meaningful, not a simple optimiser swap__.\
> The optimisation rule in test-time training defines how the fast weights evolve, and thus defines the effective inference-time model of the TTT layer. Changing this rule fundamentally alters the TTT layer’s behaviour, even if the outer-layer architecture remains similar (as has been pointed out by prior works such as LaCT [a]).
> Our introduction of Hessian-free (HF) second-order optimisation is non-trivial for several reasons:
>   - Common second-order optimisers cannot be applied because their matrix inversion steps break the gradient flow required in TTT.
>   - HF avoids explicit inversion by using conjugate gradient (CG), making it compatible with TTT while enabling higher-quality updates. To our knowledge, this is the first integration of HF optimisation with TTT.
>   - Implementing HF in TTT requires several new design decisions (Appendix A.1):
>  • choice of curvature matrix,
>  • per-token learning rates,
>  • early-stopping for parallel CG iterations.
>
> These components materially change how the memory is updated during streaming inference, resulting in consistently better performance across all benchmarks.
>
> 2. __The memory reader is redesigned for long video streaming__.
> While inspired by AdaReTaKe, our memory reader is adapted specifically for long-duration multimodal streaming, introducing:
> - a global Top-K retrieval budget rather than layer/time-specific budgets,
> - a design that maintains constant memory size regardless of video length,
>
> Empirically, this enables stable scaling to hour-long videos at 1FPS and 360p, which prior memory-reader designs do not support. As a result, the combined contribution, including TTT-HF and memory reader, improved the model performance from 45.3% and 46.9% (merging baseline) to 52.8% and 55.8% on LVBench and VideoEvalPro, two extremely long benchmarks.
>
> 3. __Summary of key contributions__
> - video-SALMONN S is the first audio-visual streaming LLM to handle hours-long 1FPS 360p videos.
> - We are the first to use TTT as a memory module for long-video multimodal understanding.
> - Our HF second-order TTT provides substantially improved memory updates and yields consistent gains across all benchmarks.
>
> > Weakness 2: Missing/insignificant quantitative comparisons - While I understand the proposed method is specifically tailored for streaming long videos, the authors do not provide a thorough comparison against non-streaming / long video understanding VLMs. For example, [2, 3] outperform almost every baseline (including video-SALMONN S) presented in this paper on LVBench, has competitive performance on MLVU, and is not included in the paper. Why are the improvements on LVBench negligible?
>
> The models in [2,3] are evaluated on LongVideoBench, whose videos are only a few minutes long. In contrast, we use LVBench, which is a different benchmark containing 3–4 hour videos and is designed to test extreme long-context understanding. Our method specifically targets this challenging setting, where prior non-streaming VLMs are not applicable.
>
> MLVU is also a short-video benchmark (~10 minutes). Even there, our streaming model achieves 73.2, outperforming LLaVA-Video’s 70.8 in non-streaming mode. The strongest non-streaming baseline, video-SALMONN 2, is already compared in Table 1.
>
> Thus, differences on LVBench reflect the difficulty of ultra-long videos, not limited effectiveness of our memory module.

---

> > ### Author Response · Authors · 2025-11-19
> > **Response to Reviewer U5i5 (part 2)**
> >
> > > Moreover, v-SALMONN-2 already appears to have competitive performance with v-SALMONN S across all datasets besides Video-MME, calling into question how much of an improvement the streaming aspect provides in long video understanding. This could potentially stem from long video datasets being unfit for truly multi-hour-long video streaming benchmarking.
> >
> > 1. Streaming models inherently operate under stricter constraints than offline models (unknown video length, fixed high frame rate), so some performance gap in shorter videos is expected.
> > 2. Even so, video-SALMONN S outperforms the strongest non-streaming baseline (video-SALMONN 2) across all benchmarks.
> > 3. To our knowledge, video-SALMONN S is the strongest streaming video understanding model for videos of more than 1 hour long, and the only audio-visual streaming model capable of handling such durations.
> >
> > > On a similar note, some presented results show that improvements are insignificant. For example, Table 3 shows that the optimizer used in the TTT layer doesn't seem to really have any significant impact on the results (SGD vs. HF), yet is the only change the authors propose for the TTT layer and is discussed in detail in Section 3.2.\
> > > [1] Dalal, K., Koceja, D., Xu, J., Zhao, Y., Han, S., Cheung, K. C., ... & Wang, X. (2025). One-minute video generation with test-time training. In Proceedings of the Computer Vision and Pattern Recognition Conference (pp. 17702-17711).\
> > > [2] Chen, Y., Xue, F., Li, D., Hu, Q., Zhu, L., Li, X., ... & Han, S. LongVILA: Scaling Long-Context Visual Language Models for Long Videos. In The Thirteenth International Conference on Learning Representations.\
> > > [3] Zhang, Y., Wu, J., Li, W., Li, B., Ma, Z., Liu, Z., & Li, C. (2024). Video Instruction Tuning With Synthetic Data. In Transactions on Machine Learning Research
> >
> > 1. __Our main improvement is not limited to optimiser choice alone__. The baseline in our study is the token merging method, and our contribution includes both (i) integrating a TTT memory module and (ii) improving its update rule. As shown in Table 2, adding the TTT layer already brings clear gains over merging alone. This directly addresses long-term memory limitations and is a core part of our contribution.
> > 2. __The effect of TTT-HF is consistent and statistically meaningful__. Across all benchmarks in Table 3, TTT-HF outperforms TTT-SGD. While individual numerical gains may appear modest, they are consistent across every dataset and are statistically significant under a __one-tailed sign test (p < 0.05)__. In the context of long-video benchmarks, where improvements are often incremental and hard to obtain, this consistency demonstrates the value of the proposed optimiser for strengthening test-time memory updates.
> >
> > We also include the suggested papers in our results section in the revised paper.
> >
> > > Question 2: Why are certain non-streaming model comparisons (see Weaknesses) left out? Either include all of them as comparisons, or discuss/exhibit why they are unfit for long-video understanding and how v-SALMONN S outperforms them.
> >
> > In addition to our explanation under Weakness 2, we clarify why certain non-streaming models are not included and how our method relates to them:
> > 1. __Long-video understanding is inherently difficult, and streaming makes it even harder__. Processing a 3-hour video at 1FPS produces ~1M visual tokens, making long-range memory a core challenge. Many offline VLMs suggested by the reviewer are not designed to operate under such extreme context lengths or streaming constraints, which is why they are unsuitable as direct baselines.
> > 2. __Our use of TTT provides substantially better long-term memory than merging alone__. We are the first to apply test-time training as a memory module for long-video understanding, and Fig. 4 shows clear advantages over token merging in preserving information as the video length grows. Our improved TTT-HF further yields consistent gains across all benchmarks.
> > 3. __Prompt-guided memory reading allows us to scale memory while preserving streaming feasibility__. This enables state-of-the-art performance, including outperforming several offline models even though we operate in a strictly harder streaming setting.
> >
> > We will make this motivation and rationale clearer in the revised manuscript.

---

> > > ### Comment · Area_Chair_g6n7 · 2025-11-26
> > >
> > > Dear reviewer U5i5,
> > >
> > > Could you please take a look at the author's response and leave your feedback?
> > >
> > > AC

---

> ### Comment · Reviewer_U5i5 · 2025-11-27
>
> Thank you to the authors for the detailed rebuttal and clarifications. After reviewing the rebuttal, I have some additional comments and questions.
>
> ---
> ### **1. Dataset Clarification**
>
> Thank you for the clarification regarding LVU vs. LongVideoBench. My concern here is resolved, however do see points 2 and 3 as they are related.
>
> ---
>
> ### **2. Claimed Benefits of TTT-HF Over TTT-SGD**
>
> I acknowledge the authors’ point that their modifications to TTT may not be entirely trivial. However, the claim that TTT-HF significantly outperforms TTT-SGD remains difficult to accept based on the reported quantitative results. The largest reported gain from TTT-SGD to TTT-HF in Table 3 is roughly **0.3%**, which is extremely small even considering the authors' claim of the difficulty of long video understanding.
>
> The authors then state in their rebuttal that the use of TTT-HF is not their only contribution, but the addition of the prompt dependent memory reading module as well. However, Table 2 shows that TTT-SGD and TTT-SGD + PD provides significant gains only on LVBench. The same can be said about TTT-HF and TTT-HF + PD. PD merging seems to only meaningfully improve performance on LVBench, with very limited contributions on VideoMME and MLVU - see my point below.
>
> ---
>
> ### **3. On Comparisons Against Non-Streaming Methods**
>
> The rebuttal argues that streaming methods operate under a different set of constraints compared to non-streaming approaches. However, this still makes it unclear why the paper compares against non-streaming models on datasets where those models inherently have an advantage.
>
> If the authors recognize that non-streaming approaches can perform better on certain datasets - and acknowledge that these datasets are not aligned with the intended streaming constraints - then:
>
> - Why include those unfit datasets? And,
> - Why compare against non-streaming models at all, especially when the goal is to evaluate the proposed contributions under realistic streaming conditions?
>
> Moreover, v-SALMONN-2, which is a non-streaming model, is already competitive with v-SALMONN S  in Table 1. The authors claim that v-SALMONN S outperforms v-SALMONN-2, but it is only by a significant margin when using both audio and video. However, in the context of video streaming, I don't find the audio modality relevant given the fact that the audio tokens bypass TTT as stated in the caption of Figure 1.   If the contribution is meant to advance streaming long-video understanding, then comparisons should be aligned with that setting.
>
> ---
>
> While I appreciate the clarifications, there are still some concerns that remain:
>
> - the addition of TTT improves long video understanding performance, but the novelty in this paper is not TTT but the use of HF optimization. **The need and benefit of HF is not quantitatively validated nor significant**.
> - the lack of strong justification for the streaming-vs-non-streaming comparisons. Additionally, the competitive performance of v-SALMONN 2 makes the contributions of v-SALMONN S unclear.
> - the improvements of the method are not consistent across all datasets. Either some datasets are not fit for long-video understanding/streaming and thus are not appropriate, or if the datasets are a fair comparison, then the claimed benefits of the proposed streaming method in this paper are not very quantitatively significant.
>
> Thank you again for the effort invested in addressing my concerns, I will read the author's next response if provided.

---

> > ### Author Response · Authors · 2025-11-27
> > **Further Response to Reviewer U5i5**
> >
> > We thank the reviewer for engaging in the discussion and further questions. We would like to address them as follows:
> > > I acknowledge the authors’ point that their modifications to TTT may not be entirely trivial. However, the claim that TTT-HF significantly outperforms TTT-SGD remains difficult to accept based on the reported quantitative results. The largest reported gain from TTT-SGD to TTT-HF in Table 3 is roughly 0.3%, which is extremely small even considering the authors' claim of the difficulty of long video understanding.
> >
> > > The authors then state in their rebuttal that the use of TTT-HF is not their only contribution, but the addition of the prompt dependent memory reading module as well. However, Table 2 shows that TTT-SGD and TTT-SGD + PD provides significant gains only on LVBench. The same can be said about TTT-HF and TTT-HF + PD. PD merging seems to only meaningfully improve performance on LVBench, with very limited contributions on VideoMME and MLVU - see my point below.
> > 1. To the best of our knowledge, __we are the first to explore TTT for long-term (multimodal) memory in video understanding__. This is an important contribution to streaming long-video understanding.
> > 2. Given the fact that we are the first to apply TTT to streaming video understanding, __the true baseline of existing work is the similarity merging__ (memory consolidation algorithm in MovieChat and VideoLLM-online). We achieved significant gains of 2.4% on VideoMME long, 0.7% on MLVU and 1.4% on LVBench compared to them, as shown in Tables 1 and 2.
> > 3. We explained in the text (lines 408-413) that the __main improvements of PD are on long videos__. We see improvements like 58.3 to 60.8 on videoMME long split (videoMME includes short, medium, and long splits ), and 47.1 to 52.8 on LVBench, which are clear gains.
> >
> > > The rebuttal argues that streaming methods operate under a different set of constraints compared to non-streaming approaches. However, this still makes it unclear why the paper compares against non-streaming models on datasets where those models inherently have an advantage.
> >
> > > If the authors recognize that non-streaming approaches can perform better on certain datasets - and acknowledge that these datasets are not aligned with the intended streaming constraints - then:
> > > - Why include those unfit datasets? And,
> > > - Why compare against non-streaming models at all, especially when the goal is to evaluate the proposed contributions under realistic streaming conditions?
> > 1. __There are no unsuitable datasets in our evaluation__. All datasets we use are standard and fully appropriate for assessing long-term memory. Our choice is simply to evaluate them in a streaming setting, which is the focus of our work.
> > 2. __Non-streaming models serve as the correct topline reference__. In non-streaming mode, the system can attend to any part of a long video at any time, provided sufficient memory is available. This offers the best achievable performance in understanding.
> > 3. In contrast, __streaming mode fundamentally restricts access to past information__. When processing video at 1 FPS, frames from 1–2 hours earlier are no longer available and can only be recalled through the model’s learned memory. This differs fundamentally from non-streaming settings, where the system can uniformly sample and attend to frames across the entire video. In practice, streaming mode is a __feasible way to handle extremely long videos under realistic memory constraints__. One may even argue that streaming is what makes human long-video understanding efficient, as humans also rely on dynamic memory rather than storing all past visual input.
> > 4. Therefore, comparing to non-streaming systems provides a meaningful upper bound, which demonstrates how well a model could perform if it had equal attention to all video content. This comparison is also standard practice in prior streaming-video research, including StreamForest [1], Dispider [2], and StreamMem [3].
> >
> > [1]. X. Zeng et al. "StreamForest: Efficient Online Video Understanding with Persistent Event Memory", 2025\
> > [2]. R. Qian et al. "Dispider: Enabling Video LLMs with Active Real-Time Interaction via Disentangled Perception, Decision, and Reaction", 2025\
> > [3]. Y. Yang et al. "StreamMem: Query-Agnostic KV Cache Memory for Streaming Video Understanding", 2025

---

> > > ### Author Response · Authors · 2025-11-27
> > > **Further Response to Reviewer U5i5 (part 2)**
> > >
> > > > Moreover, v-SALMONN-2, which is a non-streaming model, is already competitive with v-SALMONN S in Table 1. The authors claim that v-SALMONN S outperforms v-SALMONN-2, but it is only by a significant margin when using both audio and video. However, in the context of video streaming, I don't find the audio modality relevant given the fact that the audio tokens bypass TTT as stated in the caption of Figure 1. If the contribution is meant to advance streaming long-video understanding, then comparisons should be aligned with that setting.
> > >
> > > 1. We would like to clarify that video-SALMONN 2 is an audio-visual model. The visual-only version of video-SALMONN 2 is Qwen-2.5-VL SFT (row 2 of table 1). Compared to this model, our visual only video-SALMONN S achieved __67.5% to 69.3% on video-MME, 70.2% to 73.2% on MLVU, 43.6% to 52.8% on LVBench, 47.8% to 55.8% on VideoEvalPro__.
> > > 2. We did not pass the audio tokens through TTT because audio tokens are much shorter compared to visual tokens (less than 1/100 at 1FPS). Including those in TTT will cause memory degradation for visual tokens. We provide results with audio tokens also passing through TTT as follows.
> > > | Model | VMME (short/medium/long) | MLVU | LVBench |
> > > | ------ | ------ | ------ | ------ |
> > > | video-SALMONN S (TTT for V, PD) | 74.2 (80.3/74.4/67.8) | 73.1 | 51.5 |
> > > | video-SALMONN S (TTT for A+V, PD) | 73.4 (80.2/73.7/66.3) | 73.0 | 51.2 |
> > >
> > > The main degradation is on video-MME long split due to influence from audio tokens.

---

### Official Review · Reviewer_Axud · 2025-10-31

**Soundness:** 2
**Presentation:** 2
**Contribution:** 2
**Rating:** 2
**Confidence:** 3

**Summary:**

This paper proposes a method to address the memory bottleneck in long video sequence processing by applying the TTT concept to video streams. As a result, the network shows adaptive performance improvement during test time even with longer frame sequences, and is capable of processing videos of over 10K frames within a fixed memory budget. The method also features a Hessian-Free optimizer that enables efficient updates during testing.

**Strengths:**

- The model can handle long video sequences about 10K frames, mitigating the scalability limitation in long-term sequence modeling.
- The application of TTT to the video streaming environment is interesting, as it enables adaptive behavior at test time.

**Weaknesses:**

- The paper claims to preserve information via TTTHF instead of hard drop, but in practice, a cosine similarity based token discard step still exists. Because of this, the approach cannot be considered entirely different from traditional token-dropping methods, and the empirical evidence for actual information preservation remains unclear.
- Training is limited to a maximum of 1024 frames, and there is no quantitative analysis of memory efficiency (frame count vs memory usage) during training. The memory aspect thus remains insufficiently analyzed.
- The TTT concept has already been explored in prior video research (Test-Time Training on Video Streams), and similar adaptive mechanisms exist in RNN and Mamba architectures through state updates. It should be clarified whether the proposed approach's distinction is limited to the HF-based second-order optimization, or if there are structural differences beyond that.
- Experimental comparison with existing sequence compression approaches is lacking. For example, methods such as VideoLLM-online or Q-former could have been included for a fair evaluation against non-TTT token-reduction techniques, making it difficult to judge whether the proposed method is truly more effective.

**Questions:**

Please refer the weaknesses part.

---

> ### Author Response · Authors · 2025-11-19
> **Response to Reviewer Axud**
>
> We appreciate the constructive suggestions from the reviewer, and would like to address each individual concerns as follows:
>
> > Weakness 1: The paper claims to preserve information via TTTHF instead of hard drop, but in practice, a cosine similarity based token discard step still exists. Because of this, the approach cannot be considered entirely different from traditional token-dropping methods, and the empirical evidence for actual information preservation remains unclear.
>
> Thank you for raising this point. We clarify the distinction and provide supporting evidence below:\
> 1. __Our method is fundamentally different from traditional token dropping__. In our pipeline, TTT-HF is applied before the cosine-similarity-based reduction step, meaning that the information from all tokens (including those later discarded) has already been integrated into the surviving tokens through high-order fast-weight updates. Thus, unlike hard dropping methods, discarded tokens do not result in lost information; their content is already embedded in the updated memory representation.
> 2. __We provide empirical evidence that long-term information is preserved__. As shown in Fig. 4(a), when we increase the number of input frames (i.e., increase available information), merging or naive dropping methods saturate or degrade, indicating their inability to retain long-range semantics. In contrast, TTT-HF continues to improve with more frames, demonstrating that information from earlier tokens is effectively preserved after the reduction step.
>
> > Weakness 2: Training is limited to a maximum of 1024 frames, and there is no quantitative analysis of memory efficiency (frame count vs memory usage) during training. The memory aspect thus remains insufficiently analyzed.
>
> Thank you for the question. We clarify the intent and memory behaviour of our method below:\
> 1. __Our goal is to improve test-time memory quality, not training-time GPU memory efficiency__. In this paper, “memory” refers to the model’s long-term video content memory. The purpose of TTT-HF is to enhance how the model stores and updates semantic information over long video streams during inference, rather than to reduce GPU memory usage during training.
> 2. __The test-time GPU memory footprint is constant, independent of video length__. Our design maintains a fixed-size memory representation and processes frames in a streaming manner. As a result, the GPU memory required at test time does not scale with the number of frames, even when evaluating hour-long videos.
>
> We have clarified these points in the revised manuscript Section 5.1.

---

> > ### Author Response · Authors · 2025-11-19
> > **Response to Reviewer Axud (part 2)**
> >
> > > Weakness 3: The TTT concept has already been explored in prior video research (Test-Time Training on Video Streams), and similar adaptive mechanisms exist in RNN and Mamba architectures through state updates. It should be clarified whether the proposed approach's distinction is limited to the HF-based second-order optimization, or if there are structural differences beyond that.
> >
> > Thank you for the question. We clarify the distinctions between our method and prior TTT or state-update approaches as follows, and include this paper in the related work section in the revised paper:
> > 1. __Fundamentally different task setting and objective__. "Test-Time Training on Video Streams" focuses on adaptation to local frame changes at test-time and does not require long-term semantic dependencies. In contrast, our work targets multimodal-LLM-based long-video understanding, which requires accumulating, preserving, and retrieving semantic information across thousands of frames. This key difference in task directly motivates the architectural and algorithmic differences in our approach.
> > 2. __Different architecture and adaptation targets__.  "Test-Time Training on Video Streams"  updates the entire model at test time. We update only a lightweight TTT layer, specifically designed for efficient long-term multimodal memory in hour-long streaming at 1FPS.
> > 3. __State-update mechanisms (e.g., RNN/Mamba) are insufficient for long-term multimodal memory__. As shown in Table 2, state updates (i.e., Mamba) underperform parameter updates (i.e., TTT) in long-video understanding. This empirically demonstrates that high-level multimodal reasoning benefits from parameter-level adaptation rather than only recurrent hidden-state updates.
> > 4. __The contribution is not merely adding HF; it fundamentally changes the inference-time model__.   The meta-update rule defines the effective inference-time behaviour of the TTT layer. Replacing SGD with Hessian-free (HF) changes this behaviour which cannot be achieved with other more standard second-order optimisers which uses explicit matrix inversion. HF avoids inversion via CG and requires non-trivial design choices (curvature selection, per-token LR, CG early stopping). To our knowledge, this is the first integration of HF with TTT.
> >
> > __Summary of key contributions__:
> > 1. video-SALMONN S is the first audio–visual streaming LLM handling hour-long 1FPS 360p videos.
> > 2. We are the first to use TTT as a memory module for long-video multimodal understanding.
> > 3. Our HF-based second-order TTT provides significantly stronger memory updates and consistent performance improvements across all benchmarks.
> >
> > > Weakness 4: Experimental comparison with existing sequence compression approaches is lacking. For example, methods such as VideoLLM-online or Q-former could have been included for a fair evaluation against non-TTT token-reduction techniques, making it difficult to judge whether the proposed method is truly more effective.
> >
> > Thank you for the suggestion. We clarify how our baselines relate to existing sequence-compression approaches:\
> > 1. __Our merging baseline is equivalent to a state-of-the-art compression method__. The merging baseline used in our paper includes trainable projection layers, making it functionally aligned with the method proposed in Li et al. (2025) [a], which is a state-of-the-art video token-compression technique.
> > 2. __VideoLLM-online uses average spatial token pooling, which is essentially the same operation as our merging baseline (minus the learned projection)__. Therefore, its behaviour is already covered by the merging baseline we evaluate. Empirically, we also find the merging baseline to be strong and competitive across our benchmarks.
> > 3. __Q-Former is unsuitable for streaming video__. It relies on a __fixed compression ratio (e.g., K learnable queries)__. In a streaming setting, where video length grows with time, Q-Former would cause the number of tokens to grow linearly with video duration, making inference increasingly expensive. Ultimately, additional merging/discarding would still be required, so it does not address the long-term memory challenges we target.
> >
> > We have included this discussion in the revised manuscript for clarity in Section 4.2.
> >
> > [a]. Y. Li et al. "Improving llm video understanding with 16 frames per second", 2025.

---

> > > ### Comment · Reviewer_Axud · 2025-11-25
> > >
> > > I acknowledge that the authors have demonstrated performance improvements with increasing frame counts when applying TTT-HF, as shown in Figure 4. However, this merely highlights a performance differentiation against the authors' self-defined baseline. It does not serve as direct evidence to support the claim that existing methodologies, such as MovieChat or LongVILA, suffer from the severe information loss claimed by the authors. In other words, superiority over a proxy baseline does not logically lead to the conclusion of "no information loss." Without direct comparisons to these specific algorithms, the effectivity of the proposed method cannot be considered fully validated.
> > >
> > > Although the Abstract specifies a "fixed memory budget" the paper does not provide concrete details about the actual hardware memory cost. The streaming-time memory budget (peak and average VRAM) should be disclosed, and comparisons with existing methods should be conducted under the same VRAM budget for fairness. In particular, a one-to-one comparison with the Qwen2.5-VL model, which can regulate memory usage through variable resolution, would more convincingly demonstrate the utility of the proposed approach.
> > >
> > > The authors stated in the main text (Lines 369-370 of the revised version, Nov 19, 2025) that Video-LLaMA 3 was finetuned to enable audio processing. However, the scores reported in Table 1 (VideoMME 66.2, MLVU 73.0, LVBench 45.3) match the scores from the original VideoLLaMA 3 paper exactly. I request a verification of whether the data has been correctly recorded across the board.
> > >
> > > In particular, without a justified comparison of memory usage and performance improvement between PI TTT_HF, PD TTT_HF, and existing methods, it is difficult to accurately assess the intrinsic performance gain of TTT_HF.

---

> > > > ### Author Response · Authors · 2025-11-26
> > > > **Further Response**
> > > >
> > > > We appreciate the reviewer’s careful reading and would like to clarify the following points:
> > > >
> > > > > I acknowledge that the authors have demonstrated performance improvements with increasing frame counts when applying TTT-HF, as shown in Figure 4. However, this merely highlights a performance differentiation against the authors' self-defined baseline. It does not serve as direct evidence to support the claim that existing methodologies, such as MovieChat or LongVILA, suffer from the severe information loss claimed by the authors. In other words, superiority over a proxy baseline does not logically lead to the conclusion of "no information loss." Without direct comparisons to these specific algorithms, the effectivity of the proposed method cannot be considered fully validated.
> > > >
> > > > __1. Our “merging baseline” is exactly the MovieChat memory-consolidation algorithm.__
> > > >
> > > > MovieChat’s original model is nearly two years old and cannot be fairly compared with video-SALMONN S. To ensure fairness, we took the exact memory-consolidation mechanism described in MovieChat, retrained it on the same dataset and with the same base model as video-SALMONN-S, and used this as our merging baseline. This is the standard practice adopted in prior work (e.g., [1]). Thus, our baseline is a direct implementation of MovieChat’s method, not a different one invented by us.
> > > >
> > > > __2. LongVILA is not a streaming model and therefore not a suitable point of comparison.__
> > > >
> > > > We do not claim that LongVILA suffers from information loss; instead, it operates in a fundamentally different setting. LongVILA expands the context window via multi-GPU offline processing, whereas our work focuses on online/streaming video understanding. Because LongVILA cannot process video incrementally, it is unsuitable for scenarios that require real-time perception, such as embodied AI agents, while video-SALMONN-S is explicitly designed for these streaming use cases. For this reason, we compare our method against other streaming approaches rather than offline models like LongVILA.
> > > >
> > > > __3. We never claim “no information loss.”__
> > > >
> > > > Throughout the paper, we consistently describe our method as reducing or mitigating information loss, not eliminating it. Our claims are limited and aligned with empirical evidence.
> > > > In summary, our evaluation compares video-SALMONN-S directly against a fair and faithful reproduction of the MovieChat memory-consolidation algorithm (also used in VideoLLM-online). Given this alignment and the scope of our work, we believe our methodology and claims are appropriate and not overstated.
> > > >
> > > > [1]. X. Zeng et al."StreamForest: Efficient Online Video Understanding with Persistent Event Memory"
> > > >
> > > > > Although the Abstract specifies a "fixed memory budget" the paper does not provide concrete details about the actual hardware memory cost. The streaming-time memory budget (peak and average VRAM) should be disclosed, and comparisons with existing methods should be conducted under the same VRAM budget for fairness. In particular, a one-to-one comparison with the Qwen2.5-VL model, which can regulate memory usage through variable resolution, would more convincingly demonstrate the utility of the proposed approach.
> > > >
> > > > On LVBench with 1FPS and 360p resolution, video-SALMONN S (PI) has an average 30.5GB VRAM requirement with peak VRAM 37.7GB. Our merging baseline (MovieChat on our base model with our training data) has an average VRAM 28.7GB with a peak VRAM 35.8GB. The only difference is the TTT layer parameters and state cache, which are worth another 2k memory tokens. We hence compare to 18k memory merging baseline. We also increase Qwen2.5-VL to 256 frames which have the same memory footprint compared to 10k frame streaming models.
> > > >
> > > > | Model | Video-MME | MLVU | LVBench | VRAM (Peak/Avg.) |
> > > > | ------ |  ------ |  ------ |  ------ |  ------ |
> > > > | Qwen2.5-VL 256 frames | 65.6 | 71.8 | 45.7 | (37.1GB/29.8GB) |
> > > > | Merging Baseline (18k tokens) | 67.2 | 72.3 | 46.1 | (37.6GB/30.1GB) |
> > > > | video-SALMONN S (V, PI) | 67.9 | 73.2 | 47.1 | (37.7GB/30.5GB) |
> > > >
> > > > Our interpretation of "regulate memory usage" is to adjust max frames so that Qwen2.5-VL has the same VRAM stats. We appreciate it if the reviewer can clarify or suggest if this is not their intention.
> > > >
> > > > > The authors stated in the main text (Lines 369-370 of the revised version, Nov 19, 2025) that Video-LLaMA 3 was finetuned to enable audio processing. However, the scores reported in Table 1 (VideoMME 66.2, MLVU 73.0, LVBench 45.3) match the scores from the original VideoLLaMA 3 paper exactly. I request a verification of whether the data has been correctly recorded across the board.
> > > >
> > > > We apologize for this confusion and we meant to say we include Video-LLaMA 3 as a baseline, and we include video-SALMONN 2+ which was finetuned on the same training data with 16k visual tokens. We have corrected this in the revised paper.

---

### Official Review · Reviewer_HUoj · 2025-10-31

**Soundness:** 3
**Presentation:** 3
**Contribution:** 3
**Rating:** 4
**Confidence:** 3

**Summary:**

video-SALMONN S tackles the problem of doing audio–visual understanding on very long videos (hours, not minutes) when the model’s context and memory are fixed, which normally forces you to drop frames or heavily compress tokens and lose important events. The authors propose a streaming architecture with a test-time-training (TTT) style memory module that, as video chunks arrive, updates lightweight parameters to “write” what has happened so far, so the model doesn’t have to store every past token. They further make memory retrieval conditioned on the user’s current query, so at answer time the model can “read back” only the parts of memory that are relevant, keeping the overall memory size small but still supporting long-range reasoning. With an hessian free optimization to make the online updates stable and efficient, the system can process multi-hour videos (they report over 3 hours at 1 FPS, 360p) under a fixed memory budget and still outperform both offline and other streaming baselines on long-video benchmarks.

**Strengths:**

- Due to use of fixed-size memory the approach scales to really long videos

- Question/Prompt dependent token selection means efficient use of memory to store relevant tokens

- Test-time training updates the smart memory to represent past events in a compact manner

- Conjugate gradient method of test time update allows safer, better-scaled steps, while avoiding computing the full Hessian

**Weaknesses:**

- Prior works in streaming video models that utilize memory and token merging/reduction should be evaluated. e.g. CVPR 2025's "Streaming Dense Video Captioning". Xingyi Zhou, Anurag Arnab, Shyamal Buch, Shen Yan, Austin Myers, Xuehan Xiong, Arsha Nagrani, Cordelia Schmid


- Even though its a streaming video LLM the results are primarily on offline benchmarks, it would be better to have some results on streaming specific benchmarks like:

SVBench: A Benchmark with Temporal Multi-Turn Dialogues for Streaming Video Understanding. ICLR 2025.
Zhenyu Yang, Yuhang Hu, Zemin Du, Dizhan Xue, Shengsheng Qian, Jiahong Wu, Fan Yang, Weiming Dong, Changsheng Xu

StreamingBench: Assessing the Gap for MLLMs to Achieve Streaming Video Understanding
Junming Lin, Zheng Fang, Chi Chen, Zihao Wan, Fuwen Luo, Peng Li, Yang Liu, Maosong Sun

-

**Questions:**

- Since there are some efficiency considerations for this approach, could you provide some benchmarking score in terms of frames/seconds for your approach and baselines on actual hardware used for training.

---

> ### Author Response · Authors · 2025-11-19
> **Response to Reviewer HUoj**
>
> We appreciate the positive feedbacks and would like to clarify each individual concerns raised by the reviewer as follows:
>
> > Weakness 1: Prior works in streaming video models that utilize memory and token merging/reduction should be evaluated. e.g. CVPR 2025's "Streaming Dense Video Captioning". Xingyi Zhou, Anurag Arnab, Shyamal Buch, Shen Yan, Austin Myers, Xuehan Xiong, Arsha Nagrani, Cordelia Schmid
>
> Thank you for pointing out this relevant prior work. Following your suggestion, we fine-tuned Qwen2.5-VL on our data and implemented the K-means clustering strategy described in Algorithm 1 of “Streaming Dense Video Captioning” (CVPR 2025). The results are shown below:
>
> | Models | Video-MME | MLVU | LVBench |
> | -------- | ------- | ------- | ------- |
> | Qwen2.5VL + Clustering | 66.8 | 70.5 | 43.6 |
> | video-SALMONN S (V, PI) | 67.9 | 73.2 | 47.1|
>
> This clustering-based compression performs slightly worse than our merging baseline across all benchmarks. While the original paper uses 512 clusters, we adopt the standard streaming-video setting used in current works, applying 8k or 16k clusters for fair comparison. These results have been added to Table 2 in the revised paper.
>
> > Weakness 2: Even though its a streaming video LLM the results are primarily on offline benchmarks, it would be better to have some results on streaming specific benchmarks like:
> SVBench: A Benchmark with Temporal Multi-Turn Dialogues for Streaming Video Understanding. ICLR 2025. Zhenyu Yang, Yuhang Hu, Zemin Du, Dizhan Xue, Shengsheng Qian, Jiahong Wu, Fan Yang, Weiming Dong, Changsheng Xu
> StreamingBench: Assessing the Gap for MLLMs to Achieve Streaming Video Understanding Junming Lin, Zheng Fang, Chi Chen, Zihao Wan, Fuwen Luo, Peng Li, Yang Liu, Maosong Sun
>
> Thank you for the helpful suggestions. We address them as follows:
>
> - We clarify that our method has already been evaluated on OVO-Bench, a streaming benchmark designed to test long-horizon understanding (Table 4 in Appendix B).
> - Most existing streaming benchmarks primarily evaluate short-term or recent-frame understanding, whereas our work focuses on enhancing long-term memory over extended video streams. Thus, their evaluation protocols are not fully aligned with our core objective.
> - That said, we appreciate the recommendation and have now added both SVBench and StreamingBench to the revised paper.
> Below, we report our results on StreamingBench (all videos <30 minutes, evaluated under full-context mode):
>
> | Models | Real-Time | Omni-Source | Contextual | Overall |
> | -------- | ------- | ------- | ------- | ------- |
> | Dispider | 67.6 | 35.7 | 33.6 | 53.1 |
> | ViSpeak s2 | 74.4 | __53.5__ | __39.6__ | 62.0 |
> | Qwen2.5 VL (offline, 4fps) | 75.4 | 38.7 | 32.1 | 57.6 |
> | Our Merging Baseline (online) | 75.1 | 39.1 | 33.8 | 57.9 |
> | video-SALMONN S (online) (V, PI) | 75.7 | 40.7 | 35.7 | 58.7 |
> | video-SALMONN S (online) (A+V, PI) | __76.0__ | 53.0 | 36.1 | __62.1__ |
>
> Although our method is not specifically optimised for real-time or contextual (proactive/sequential) queries, video-SALMONN S achieves performance on par with the best open-source models on StreamingBench. We have added this analysis to Section 5.1 of the revised manuscript.

---

> > ### Comment · Area_Chair_g6n7 · 2025-11-26
> >
> > Dear reviewer HUoj,
> >
> > Could you please take a look at the author's response and leave your feedback?
> >
> > AC

---

### Official Review · Reviewer_PaH2 · 2025-11-01

**Soundness:** 4
**Presentation:** 3
**Contribution:** 3
**Rating:** 6
**Confidence:** 3

**Summary:**

This paper presents Video-SALMONN S, a scalable framework for **streaming multimodal video understanding** that enables large video-language models to process arbitrarily long videos under fixed memory and computation budgets. The core innovation lies in combining Test-Time Training with Hessian-Free optimization (TTTHF)—which updates visual memory parameters during inference to retain long-term information—with a prompt-dependent memory reading mechanism that selectively retrieves task-relevant visual tokens, and a similarity-based discarding strategy to prevent unbounded KV cache growth. Extensive experiments on multiple benchmarks, including Video-MME, LVBench, MLVU, and VideoEvalPro, show that Video-SALMONN S achieves superior performance and efficiency over recent streaming baselines (e.g., StreamMem, Dispider, MovieChat), supporting up to three-hour-long videos while maintaining competitive accuracy. Overall, the paper provides a well-engineered and empirically validated solution to the scalability bottleneck of streaming video-language models.

**Strengths:**

1. The integration of Test-Time Training with Hessian-Free optimization (TTTHF) and prompt-dependent memory reading is novel and effectively addresses the long-standing problem of unbounded visual KV cache accumulation in streaming settings. This design provides a fresh perspective by merging adaptive optimization with memory-efficient retrieval, extending prior works such as MovieChat, StreamMem, and ReTaKe toward a truly long-duration streaming paradigm.

2. The experimental evaluation is comprehensive and convincing, spanning multiple long-form benchmarks (Video-MME, LVBench, MLVU, and VideoEvalPro). Results demonstrate consistent improvements in both accuracy and efficiency, supporting up to three-hour-long videos under fixed GPU memory—an impressive engineering achievement. Ablation studies thoroughly validate the necessity of each module (TTTHF, similarity-based discarding, and prompt-dependent retrieval), showing clear causal links between design choices and performance gains.

**Weaknesses:**

1. The similarity-based discarding mechanism, although efficient, inherently risks removing semantically critical frames or rare events. The paper lacks a detailed analysis of potential information loss and does not provide qualitative examples or failure cases illustrating where compression harms temporal or semantic fidelity.

2. The proposed TTTHF module introduces additional test-time optimization overhead, yet the paper provides only high-level runtime statistics without precise measurements of latency, FLOPs, or energy consumption. Moreover, details such as GPU hours, batch configurations, and training cost breakdown are missing, making it difficult to assess real-world deployability and reproducibility. The “test-time training while streaming” paradigm incurs extremely high computational costs, making it difficult—if not impossible—for the model to achieve truly real-time streaming performance.

**Questions:**

1. The similarity-based discarding strategy effectively constrains memory usage but may also risk removing rare yet semantically important frames. Could the authors provide further analysis or visualization to quantify how much semantic or temporal information is lost due to discarding?

2. TTTHF adds test-time optimization overhead, yet the paper reports only aggregate runtime metrics. Could the authors include detailed latency, FLOPs, and energy profiling under different settings, along with training cost (GPU hours, batch size, memory)? Finally, are there lightweight or approximate TTTHF variants that maintain performance while reducing computation?

---

> ### Author Response · Authors · 2025-11-19
> **Response to Reviewer PaH2**
>
> Thank you for your positive comments and constructive feedbacks. We would like to respond to each concerns as follows:
>
> > Weakness 1: The similarity-based discarding mechanism, although efficient, inherently risks removing semantically critical frames or rare events. The paper lacks a detailed analysis of potential information loss and does not provide qualitative examples or failure cases illustrating where compression harms temporal or semantic fidelity.
>
> > Question 1: The similarity-based discarding strategy effectively constrains memory usage but may also risk removing rare yet semantically important frames. Could the authors provide further analysis or visualization to quantify how much semantic or temporal information is lost due to discarding?
>
> Thank you for raising this point. We agree that naively discarding frames could risk eliminating semantically important or rare events. However, our method is specifically designed to mitigate this issue by preserving diverse semantic content through token clustering and hierarchical selection. To further clarify, Fig. 4(a) directly reflects this behaviour: as the number of frames increases, the performance of methods involving token merging or naive dropping degrades, suggesting that critical information is being lost. In contrast, TTT-HF has clearer trends to improve with more frames, showing that our strategy preserves long-term information better while maintaining computational efficiency.
>
> > Weakness 2: The proposed TTTHF module introduces additional test-time optimization overhead, yet the paper provides only high-level runtime statistics without precise measurements of latency, FLOPs, or energy consumption. Moreover, details such as GPU hours, batch configurations, and training cost breakdown are missing, making it difficult to assess real-world deployability and reproducibility. The “test-time training while streaming” paradigm incurs extremely high computational costs, making it difficult—if not impossible—for the model to achieve truly real-time streaming performance.
>
> > Question 2: TTTHF adds test-time optimization overhead, yet the paper reports only aggregate runtime metrics. Could the authors include detailed latency, FLOPs, and energy profiling under different settings, along with training cost (GPU hours, batch size, memory)? Finally, are there lightweight or approximate TTTHF variants that maintain performance while reducing computation?
>
> Thank you for raising this concern. We have now added detailed latency, FLOPs, and runtime profiling to the revised paper, along with training-cost clarification. In summary, the additional computation introduced by TTT-HF is minimal relative to the LLM backbone, does not affect real-time performance, and is further stabilized by our high-order optimisation. TTT-HF enables dynamic memory updates during streaming, which is crucial for long-video understanding and significantly improves efficiency and robustness in practical deployments.
>
> __Details of the computation__:
> For a TTT-layer with input dimension $d$, intermediate dimension $m$, total sequence length $T$, minibatch size $B$ and minibatch size $B$, all three methods share the same core operations:
> For all the methods: TTTSGD, TTTMuon, TTTHF, they all require the following common operations:
>   - Forward operation with pre-updated fast weight ($2\times Tdm$)
>   - Backward operation to compute the gradient ($4\times Tdm$)
>   - Meta-optimiser-specific update generation cost (different across methods)
>   - Final forward operation for inference ($2\times Tdm$)
>
> This yields a constant base cost $8\times Tdm$ for all variants. Extra costs caused by different optimisers are:
>   - SGD: no extra FLOPs for generating update.
>   - Muon: ~5 NS iterations, each with 3 parameters, which in total costs $T\times (10\times dm + 20\times d^2)\times (d/B)$
>   - TTT-HF: ~3 matrix vector products with the curvature matrix, which in total costs $18\times Tdm$
> Overall, TTT-HF has competitive computation complexity as compared to TTT-Muon (depending on the ratio of $d/B$, HF becomes more efficient than Muon for a smaller $B$). Overall, TTT-HF costs around ~2.25x more when compared to TTT-SGD, but since the TTT layer is a single small MLP applied only to input tokens, the added FLOPs are negligible relative to the LLM backbone’s inference cost.
>
> We provide our test-time runtime metrics as follows (measured across LVBench):
> | Module | Prefill TFLOPs (per 1024 tokens) | Inference Time per Frame (s) | First Token Latency (s)    |
> | -------- | ------- | ------- | ------- |
> | Qwen2.5 VL + Merge | 2.76 | 0.00398 | 0.799 |
> | + TTT-SGD | 2.76+0.000206 | 0.00409 | 0.802 |
> | + TTT-HF (3 steps) | 2.76+0.000668 | 0.00436 | 0.805 |

---

> > ### Author Response · Authors · 2025-11-19
> > **Response to Reviewer PaH2 (Part 2)**
> >
> > __Key observations:__
> >   1. TFLOPs overhead is negligible compared to the entire LLM because it is just a single MLP.
> >   2. Inference time per frame overhead is negligible compared to the first token latency.
> >   3. TTT-HF does not influence first token latency because it updates memory whenever new frames come. So when a user prompt comes, the visual memory is always ready to be used by the LLM.
> >
> > As a result, with TTT-HF, for 1-hour video at 1FPS and 360p, we can still process within 20 seconds on average. We added this table to the revised paper.
> >   - We have now added these to the revised paper.
> >   - We discussed training cost (GPU hours) in Section 4.3. We added in the revised paper that we use a batch size of 1 per GPU. We will open-source training data and configurations to ensure reproducibility.
> >
> > Overall, test-time training for memory updates provides a lightweight mechanism to dynamically reshape the memory network during streaming, which analogous to how human memory mechanism continuously reorganises neural connections when processing new information. This capability is crucial for making long-video understanding more efficient in real deployment.
> >
> > Although TTT adds computation within the memory layer, the overhead is negligible compared to the total cost of the LLM backbone and does not hinder real-time performance. Test-time training has already proven feasible in practical systems (e.g., long-video generation, reinforcement learning). While SGD training for the TTT-layer may introduce instability, our TTT-HF module addresses this challenge via stable high-order optimisation, ensuring both robustness and efficiency.

---

> > > ### Comment · Area_Chair_g6n7 · 2025-11-26
> > >
> > > Dear reviewer PaH2,
> > >
> > > Could you please take a look at the author's response and leave your feedback?
> > >
> > > AC

---

### Author Response · Authors · 2025-12-03

Dear Area Chair,

We sincerely appreciate the reviewers for the time and effort they dedicated to evaluating our work, and we also thank the Area Chairs and Program Chairs for their attentive management of the review process.

Unfortunately, we found that several reviewers (HUoj, Axud, U5i5) have significant misunderstanding of key elements in our paper, and hence giving misleading questions resulted in unexpectedly low evaluations. Below, we restate our contributions and address the main points of confusion.

### Main Contributions:
1. video-SALMONN S is __the first audio–visual streaming LLM__ that can handle more than 3-hour-long videos at 1FPS and 360P resolution, achieving state-of-the-art performance on longc-video benchmarks such as video-MME.
2. video-SALMONN S is __the first to explore TTT for long-term (multimodal) memory in video understanding__. This is an important contribution to streaming long-video understanding.
3. Our HF-based second-order TTT is well-motivated and provides __significantly stronger memory updates and consistent performance improvements across all benchmarks__.

### Clarification of Misunderstandings:
1. Reviewer __HUoj__ requested clustering baseline and results on StreamingBench, which have been __fully__ addressed in our rebuttal, but have not yet received acknowledgements from the reviewer. Although real-time perception is not the focus of this paper, video-SALMONN S still achieved __state-of-the-art__ performance on StreamingBench across open-source models.
2. Reviewer __Axud__ had key misunderstandings regarding our contribution and baselines:
  - The "Test-Time Training on Video Streams" that reviewer Axud pointed out has __fundamentally__ different objectives, task settings, and update mechanisms. Therefore, the reviewer completely __ignored that we are the first__ to explore TTT for long-term memory in video understanding.
  - The MovieChat baseline the reviewer mentioned is __already included in our paper__ by implementing its core memory consolidation mechanism (referred to as the merging baseline in the paper). We also compared a __stronger__ compression method, F16, with a trainable merging layer. The LongVILA, the reviewer suggested, is for offline models and suffers from high latency, hence __not suitable for comparison__.
  -  We have provided __all the comparisons and memory usage__ requested by the reviewer.
3. Reviewer __U5i5__ had key misunderstandings regarding contribution and baselines:
  - The reviewer misunderstood our use of non-streaming models. Like many other papers, we use non-streaming models as a __correct topline reference__ which demonstrates how well a model could perform if it had equal attention to all video content.
  - The reviewer __ignored the fact that we are the first__ to explore TTT for long-term memory in video understanding, and no one has ever done this exploration before.
  - The reviewer overlooked that video-SALMONN 2 is an audio-visual model, and the visual-only version of video-SALMONN S should be compared to Qwen-2.5-VL SFT (row 2 of table 1). In visual-only model comparison, we achieved improvements of __67.5% to 69.3% on video-MME, 70.2% to 73.2% on MLVU, 43.6% to 52.8% on LVBench, and 47.8% to 55.8% on VideoEvalPro. These are large improvements__.
4. __Reviewer PaH2 had no substantial issue, and__ we provided the run-time statistics, including TFLOPS and latency, in response, which resolved the concerns.

Given the significant misunderstandings in the reviews, we respectfully request that the Area Chair reevaluate our submission in light of its contributions and our comprehensive rebuttal.

Thank you again for your consideration.

Best regards, The Authors

---

### Meta-Review · Area_Chair_SSv9 · 2026-01-07

**Summary:**

## Summary
The paper proposes video-SALMONN-S, which the author(s) claimed to be the *first audio–visual streaming LLM* and also be the first *the first to explore TTT for long-term (multimodal) memory in video understanding*. On the reviewers concerns, two (Axud , U5i5) concerned about the limitted novelty and / or limitied technical contributions. Other concerns include missing comparions and unconvincing results.

## Discussions & Decision
AC reads all reviews from the reviewers and their interaction with the authors. While it is true that video-SALMONN-S is "the first to explore TTT for long-term (multimodal) memory in video understanding", it is also a correct statement from the reviewer Axud that "TTT concept has already been explored in prior video research" as in [1] for video understanding and [2] for video generation. The author(s) are true that the problem setup is different, for video-SALMONN-S, it is long-term (multimodal) memory in video understanding. However,  the *small* problem difference (video understanding vs. long-term multimodal memory in video understanding) does not make a significantly novel contribution and, in the other hand, the paper does not provide solid enough experimental results to convince the reivewers that strong results can outweigh the lack of novelty. AC believes that the paper has the potential. However, it may need more revisions to solidify its experiments to be more conving to reviewers and readers. AC recommends to reject this paper in its current form and encourages the author(s) improve their paper and re-submit it to future conferences.

[1] Test-Time Training on Video Streams, JMLR'25

[2] One-Minute Video Generation with Test-Time Training, CVPR'25

**Reviewer Concerns:**

* Heavy reliance on prior work/limited novelty (Axud , U5i5)
* Missing some baseline comparisons (Axud, U5i5, HUoj)
* Unconving comparison / aka improvement over author self-defined baselines (Axud)

**Reviewer Scores:**

The paper initially receives scores of 6, 4, 2, 4 from four different reviewers. After rebuttal, most of the reviewers keep their opinion unchanged.

---

### Decision · Program_Chairs · 2026-01-26

Reject